# Non-Stationary Learning of Neural Networks with Automatic Soft Parameter Reset

**Alexandre Galashov**[*]
Gatsby Unit, UCL
Google DeepMind
agalashov@google.com

**Michalis K. Titsias**
Google DeepMind
mtitsias@google.com

**András György**
Google DeepMind
agyorgy@google.com

**Clare Lyle**
Google DeepMind
clarelyle@google.com

**Razvan Pascanu**
Google DeepMind
razp@google.com

**Yee Whye Teh**
Google DeepMind
University of Oxford
ywteh@google.com

**Maneesh Sahani**
Gatsby Unit, UCL
maneesh@gatsby.ucl.ac.uk

## Abstract

Neural networks are most often trained under the assumption that data come from a stationary distribution. However, settings in which this assumption is violated are of increasing importance; examples include supervised learning with distributional shifts, reinforcement learning, continual learning and non-stationary contextual bandits. Here, we introduce a novel learning approach that automatically models and adapts to non-stationarity by linking parameters through an Ornstein-Uhlenbeck process with an adaptive drift parameter. The adaptive drift draws the parameters towards the distribution used at initialisation, so the approach can be understood as a form of *soft* parameter reset. We show empirically that our approach performs well in non-stationary supervised, and off-policy reinforcement learning settings.

## 1 Introduction

Neural networks (NNs) are typically trained using algorithms such as stochastic gradient descent (SGD), which implicitly assume that the data come from a stationary distribution. This assumption is incorrect for scenarios such as continual learning, reinforcement learning, non-stationary contextual bandits, and supervised learning with distribution shifts [20, 53]. Although the parameters can be updated online as new data are encountered, this approach often leads to a *loss of plasticity* [12, 2, 13], manifesting either as a failure to generalise to new data despite reduced training loss [4, 2], or as an inability to reduce training error as the data distribution changes [13, 37, 1, 42, 34].

In [38], the authors argue for two factors that lead to loss of plasticity: shifts in the distribution of preactivations, leading to dead or dormant neurons [47], and growth in the parameter norm, causing training instabilities. These problems are often addressed using *hard resets* based on heuristics such as detecting dormant units [47], assessing unit utility [13, 12], or simply after a fixed number of steps has elapsed[43]. Although effective at increasing plasticity, hard resets can be inefficient as they can discard valuable knowledge captured by the parameters.

---

[*]Corresponding author

38th Conference on Neural Information Processing Systems (NeurIPS 2024).

We propose an algorithm that instead implements a form of *soft* parameter reset, avoiding the pitfalls associated with hard resets. A *soft* reset moves the parameters in the direction of the initial-value distribution, while maintaining dependence on their previous values. It also increases the learning rate applied to gradient-based updates, allowing new parameters to adapt faster to the changing data. The magnitude of the soft reset is governed by an Ornstein-Uhlenbeck *drift* process linking the parameters in time, with the scale of the drift itself chosen adaptively. In effect, the model approximates the action of a dynamical Bayesian prior over the NN parameters, which is adapted online to new data.

Our contributions can be summarised as follows. We provide an explicit formalisation of the model for the drift in NN parameters as sketched above, and derive a procedure to estimate the parameters of the drift model online. Second, we use the learnt drift model to modify the NN parameter update algorithm. Third, we explore the effectiveness of the approach in supervised learning experiments, showing that it avoids the challenge of plasticity loss, as well as in an off-policy reinforcement learning setting.

## 2 Non-stationary learning with Online SGD

Consider a non-stationary supervised learning setting with changing data distributions $p_t(x, y)$, where $x \in \mathbb{R}^L$ and $y \in \mathbb{R}^K$ ($y$ may be restricted to "one-hot" categorical indicator vectors). Given a single-point loss $\mathcal{L}(\theta, x, y)$, we define the time-dependent expected loss function for parameters $\theta \in \mathbb{R}^D$ to be

$$\mathcal{L}_t(\theta) = \mathbb{E}_{(x_t, y_t) \sim p_t} \mathcal{L}(\theta, x_t, y_t) . \tag{1}$$

Our goal is to find a parameter sequence $\Theta = (\theta_1, \ldots, \theta_T)$ that reduces the dynamic regret

$$R_T(\Theta, \Theta^\star) = \tfrac{1}{T} \sum_{t=1}^T \left( \mathcal{L}_t(\theta_t) - \mathcal{L}_t(\theta_t^\star) \right), \tag{2}$$

relative to an oracular reference sequence $\Theta^\star = (\theta_1^\star, \ldots, \theta_T^\star)$ satisfying $\theta_t^\star \in \arg\min_\theta \mathcal{L}_t(\theta)$. One common approach to online learning employs online stochastic gradient descent (SGD) [23]. An initial parameter value $\theta_0$ is updated sequentially for each batch of data $\mathcal{B}_t = \{(x_t^i, y_t^i)\}_{i=1}^B$ s.t. $(x_t^i, y_t^i) \sim p_t(x_t, y_t)$. The update rule is:

$$\theta_t = \theta_{t-1} - \alpha_t \nabla_\theta \widehat{\mathcal{L}}_t(\theta_{t-1}),$$

where $\widehat{\mathcal{L}}_t(\theta) = \frac{1}{B} \sum_{i=1}^B \mathcal{L}(\theta, x_t^i, y_t^i)$ is the empirical loss on the batch, and $\alpha_t$ is a learning rate. See also Appendix G for the connection of SGD to proximal optimization.

**Convex settings.** For convex losses, online SGD with a fixed learning rate $\alpha$ can track non-stationarity [56]. By selecting $\alpha$ appropriately—potentially using additional knowledge about the reference sequence—we can optimise the dynamic regret in (2). In general, algorithms that adapt to the observed level of non-stationarity can outperform standard online SGD. For example, in [29], the authors propose to adjust the learning rate $\alpha_t$, while in [21] and in [29], the authors suggest modifying the starting point of SGD from $\theta_t$ to an adjusted $\theta_t'$ proportional to the level of non-stationarity.

**Non-convex settings.** Non-stationary learning with NNs is more complex, since now there is a changing set of local minima as the data distribution changes. Such changes can lead to a loss of plasticity and other pathologies. Alternative optimization methods like Adam [30], do not fully resolve this issue [13, 37, 1, 42, 34]. Parameter resets [13, 48, 12] partially mitigate the problem, but may be too aggressive if the data distributions are similar.

## 3 Online non-stationary learning with learned soft parameter resets

**Notation.** We denote by $\mathcal{N}(\theta; \mu, \sigma^2)$ a Gaussian distribution on $\theta$ with mean $\mu$ and variance $\sigma^2$. We denote by $\theta^j$ the $j$-th component of the vector $\theta = (\theta^1, \ldots, \theta^D)$. We assume that the NN outputs define a distribution over targets $y$, so that the single-point loss is the negative log likelihood $\mathcal{L}(\theta, x, y) = -\log p(y|x, \theta)$. As above, $\mathcal{B}_t = \{(x_t^i, y_t^i)\}_{i=1}^B$ is a batch of data sampled from $p_t(x, y)$ with a fixed batch size $B$ (which could be 1) and $\widehat{\mathcal{L}}_t(\theta) = \log p(\mathcal{B}_t|\theta) = \frac{1}{B} \sum_{i=1}^B \mathcal{L}(\theta, x_t^i, y_t^i)$ is the average negative log likelihood on batch $\mathcal{B}_t$. Element-wise product is denoted by $\circ$.

Our proposed *soft reset* scheme tracks changes in the distribution of non-stationary data using an explicit *drift* model for the parameters $p(\theta_{t+1}|\theta_t, \gamma_t)$. Specifically, the drift model we adopt assumes

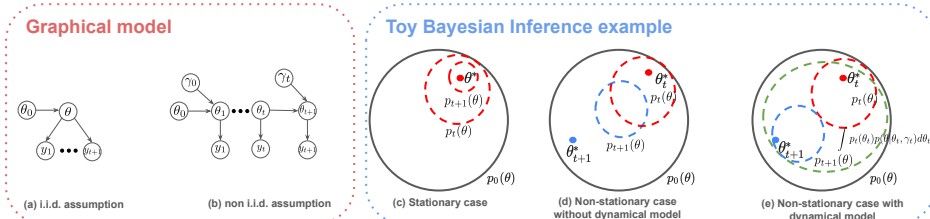

Figure 1: **Left**: graphical model for data generating process in the (a) stationary case and (b) non-stationary case with drift model $p(\theta_{t+1}|\theta_t, \gamma_t)$. **Right**: (c) In a stationary online learning regime, the Bayesian posterior (red dashed circles) in the long run will concentrate around $\theta^*$ (red dot). (d) In a non-stationary regime where the optimal parameters suddenly change from current value $\theta_t^*$ to new value $\theta_{t+1}^*$ (blue dot) online Bayesian estimation can be less data efficient and take time to recover when the change-point occurs. (e) The use of $p(\theta|\theta_t, \gamma_t)$ and the estimation of $\gamma_t$ allows to increase the uncertainty, by soft resetting the posterior to make it closer to the prior (green dashed circle), so that the updated Bayesian posterior $p_{t+1}(\theta)$ (blue dashed circle) can faster track $\theta_{t+1}^*$.

that the parameters change in the direction of the initialization distribution at each time $t$. The amount of change, and thus the strength of reset, is controlled by non-stationary hyperparameters $\gamma_t$. These hyperparameters are themselves estimated online from the data.

From a probabilistic standpoint, the drift model implies an *empirical Bayesian* prior on parameters $\theta_{t+1}$, which is displaced from the posterior on $\theta_t$ towards the prior distribution, and has increased variance (controlled by $\gamma_t$). As we argue below, the effect can also be interpreted in the context of SGD learning, as an adjustment in the starting point for batch $\mathcal{B}_{t+1}$ to a point $\tilde{\theta}_t(\gamma_t)$ between the previous estimate $\theta_t$ and the mean of initializing distribution, with a corresponding increase in the learning rate. This approach is inspired by prior work in online convex optimization for non-stationary environments [e.g., 25, 21, 8, 18, 29].

### 3.1 Toy illustration of the advantage of drift models

Consider online Bayesian inference with 2-D observations $y_t = \theta^\star + \epsilon_t$ , where $\theta^\star \in \mathbb{R}^2$ are unknown *true* parameters and $\epsilon_t \sim \mathcal{N}(0; \sigma^2 I)$ is Gaussian noise with variance $\sigma^2$. Starting from a Gaussian prior $p_0(\theta) = \mathcal{N}(\theta; \mu_0, \Sigma_0)$, the posterior distribution $p_{t+1}(\theta) = p(\theta|y_1, \ldots, y_{t+1}) = \mathcal{N}(\theta; \mu_{t+1}, \Sigma_{t+1})$ is updated using Bayes' rule

$$p_{t+1}(\theta) \propto p(y_{t+1}|\theta)p_t(\theta). \tag{3}$$

The posterior update (3) arises from assumption that data are i.i.d. (Figure 1a), since then $p_{t+1}(\theta) \propto p_0(\theta) \prod_{s=1}^{t+1} p(y_s|\theta)$. By the Central Limit Theorem, the posterior mean $\mu_t$ converges to $\theta^\star$ and the covariance matrix $\Sigma_t$ shrinks to zero (the radius of red circle in Figure 1c).

Suppose now that the *true* parameters $\theta_t^\star$ (which were fixed up to time $t$) change to new parameters $\theta_{t+1}^\star$ at time $t + 1$. The i.i.d. assumption is thus violated and the update (3) becomes problematic because the low uncertainty (small radius of red dashed circle in Figure 1d) in $p_t(\theta)$ causes the posterior $p_{t+1}(\theta)$ (see blue circle) to adjust slowly towards $\theta_{t+1}^\star$ (blue dot) as illustrated in Figure 1d.

The issue is addressed by allowing explicitly for the possibility that, before observing new data, the parameters *drift* according to $p(\theta_{t+1}|\theta_t, \gamma_t)$. The corresponding conditional independence structure is shown in Figure 1b. The posterior update then becomes:

$$p_{t+1}(\theta) \propto p(y_{t+1}|\theta) \int p(\theta|\theta_t', \gamma_t)p_t(\theta_t')d\theta_t'. \tag{4}$$

For a suitable choice of drift model $p(\theta_{t+1}|\theta_t, \gamma_t)$, this modification allows $p_{t+1}(\theta)$ (blue circle) to adjust more rapidly towards the new $\theta_{t+1}^\star$ (blue dot), see Figure 1e. This is because the new prior $\int p(\theta|\theta_t', \gamma_t)p_t(\theta_t')d\theta_t'$ has larger variance (green circle) than $p_t(\theta)$ and its mean is closer to the center of the circle. Ideally, the parameter $\gamma_t$ should capture the underlying non-stationarity in the data distribution in order to control the impact of the prior $\int p(\theta|\theta_t', \gamma_t)p_t(\theta_t')d\theta_t'$. For example, if at some point the non-stationarity disappears, we want the drift model to adaptively eliminate changes and recover the stationary posterior update (3). This highlights the importance of the adaptive nature of the drift model.

## 3.2 Ornstein-Uhlenbeck parameter drift model

We motivate a specific choice of drift model which is useful for maintaining plasticity. Assume that the NN is flexible enough to capture any *stationary* dataset in a fixed number of iterations starting from a good initialization $\theta_0 \sim p_0(\theta)$ [see, e.g., 24, 16]. Informally, we refer to the region of high probability under $p_0$ as a *plastic region*.

In a non-stationary data setting, learning for the batch $\mathcal{B}_{t+1}$ is initialized based on the learnt parameter at the previous time step, $\theta_t$. Empirically, if this parameter is far from the initial plastic region, the NN may suffer from a *loss of plasticity* with SGD and similar online methods failing to adapt to the new data. In this case, a *hard reset* of the parameters to $\theta_0$ (or an alternative draw from $p_0$) may be helpful. However, if changes in the data distribution are gradual then performance may be improved by retaining information about $\theta_t$. Furthermore, since $\theta$ is high-dimensional, each dimension might benefit from different updates.

**Drift model.** A drift model $p(\theta_{t+1}|\theta_t, \gamma_t)$ which captures these desiderata is given by

$$p(\theta|\theta_t, \gamma_t) = \prod_j \mathcal{N}(\theta^j; \gamma_t^j \theta_t^j + (1 - \gamma_t^j)\mu_0^j, (1 - \gamma_t^{j\,2})\sigma_0^{j\,2}), \tag{5}$$

defined independently for each parameter dimension $\theta^j$, where $p_0(\theta_0^j) = \mathcal{N}(\theta_0^j; \mu_0^j; \sigma_0^{j\,2})$ is the per-parameter prior distribution and $\gamma_t = (\gamma_t^1, \ldots, \gamma_t^D)$ controls drift in each parameter separately. The model is a discretized Ornstein-Uhlenbeck (OU) process [50] (see Appendix B for the derivation).

The parameters $\gamma_t^j \in [0, 1]$ control the degree to which each parameter is reset. When $\gamma_t^j = 1$, the model has $\theta_{t+1}^j$ and $\theta_t^j$ equal, and the data in $\mathcal{B}_{t+1}$ are used to refine the estimate of the common parameter. When $\gamma_t^j = 0$, $\theta_{t+1}^j$ is drawn from the prior $p_0$ independently of $\theta_t^j$, and so $\mathcal{B}_{t+1}$ is used independently of previous data. A value of $\gamma_t^j \in (0, 1)$ interpolates between these two extremes. The process (5) has the property that for any current parameter $\theta_t$ and $\gamma_t^j \in [0, 1)$, as $T \to \infty$ the distribution of $p(\theta_T|\theta_t)$ will converge to the prior $p(\theta_0)$. This behaviour depends on interplay between the mean shrinkage and the variance. Other choices of drift variance would result in the variance of $p(\theta_T|\theta_t)$ either going to $0$ or growing to $\infty$, harming learning. Thus, the model (5) encourages parameters to move towards the plastic region (initialization). In Appendix C, we discuss this further and other potential choices for the drift model.

## 3.3 Online estimation of drift

The parameters $\gamma_t$ can themselves be selected within the empirical Bayesian framework by optimizing the *predictive likelihood*, which quantifies the probability of new data under the current parameters and drift model. We derive the drift estimation procedure in the context of approximate online variational inference [7] with Bayesian Neural Networks (BNN). Let $\Gamma_t = (\gamma_1, \ldots, \gamma_t)$ be the history of observed parameters of the drift model and $\mathcal{S}_t = \{\mathcal{B}_1, \ldots, \mathcal{B}_t\}$ be the history of observed data. The objective of approximate online variational inference is to propagate an *approximate* posterior $q_t(\theta|\mathcal{S}_t, \Gamma_{t-1})$ over parameters, such that it is constrained to some family $\mathcal{Q}$ of probability distributions. In the context of BNNs, it is typical [5] to assume a family $\mathcal{Q} = \{q(\theta) : q(\theta) \sim \prod_{j=1}^{D} \mathcal{N}(\theta^j; \mu^j, \sigma^{j\,2}); \theta = (\theta^1, \ldots, \theta^D)\}$ of Gaussian mean-field distributions over parameters $\theta \in \mathbb{R}^D$ (separate Gaussian per parameter). Let $q_t(\theta) \triangleq q_t(\theta|\mathcal{S}_t, \Gamma_{t-1}) \in \mathcal{Q}$ be the Gaussian *approximate* posterior at time $t$ with mean $\mu_t$ and variance $\sigma_t^2$ for every parameter. The new approximate posterior $q_{t+1}(\theta) \in \mathcal{Q}$ is found by

$$q_{t+1}(\theta) = \arg\min_q \mathbb{KL}\left[q(\theta)||p(\mathcal{B}_{t+1}|\theta)q_t(\theta|\gamma_t)\right], \tag{6}$$

where the prior term is the approximate predictive look-ahead prior given by

$$q_t(\theta|\gamma_t) = \int q_t(\theta_t)p(\theta|\theta_t, \gamma_t)d\theta_t = \mathcal{N}(\theta; \tilde{\mu}_t(\gamma_t), \tilde{\sigma}_t^2(\gamma_t)) \tag{7}$$

that has parameters $\tilde{\mu} = (\tilde{\mu}^1, \ldots, \tilde{\mu}^D)$ and $\tilde{\sigma}^2 = (\tilde{\sigma}^{1\,2}, \ldots, \tilde{\sigma}^{D\,2})$ such that $\tilde{\mu}_t^j(\gamma_t) = \gamma_t^j \mu_t^j + (1 - \gamma_t^j)\mu_0^j$, $\tilde{\sigma}_t^{j\,2}(\gamma_t) = \gamma_t^{j\,2}\sigma_t^{j\,2} + (1 - \gamma_t^{j\,2})\sigma_0^{j\,2}$, see Appendix E.1 for derivation. The form of this prior $q_t(\theta|\gamma_t)$ comes from the non i.i.d. assumption (see Figure 1b) and the form of the drift model (5). For new batch of data $\mathcal{B}_{t+1}$ at time $t + 1$, the *approximate predictive log-likelihood* equals to

$$\log q_t(\mathcal{B}_{t+1}|\gamma_t) = \log \int p(\mathcal{B}_{t+1}|\theta)q_t(\theta|\gamma_t)d\theta. \tag{8}$$

The log-likelihood (8) allows us to quantify predictions on batch of data $\mathcal{B}_{t+1}$ given our current distribution $q_t(\theta)$ and the drift model from (5). We find such $\gamma_t^\star$ that

$$\gamma_t^\star \approx \arg\max_{\gamma_t} \log q_t(\mathcal{B}_{t+1}|\gamma_t) \tag{9}$$

Using $\gamma_t^\star$ in (5) modifies the prior distribution (7) to fit the most recent observations the best by putting more mass on the region where the new parameter could be found (see Figure 1,right).

**Gradient-based optimization for $\gamma_t$.** The approximate predictive prior in (7) is Gaussian which allows us to use the so-called reparameterisation trick to optimize (8) via gradient descent. Starting from an initial value of drift parameter $\gamma_t^0$ at time $t$, we perform $K$ updates with learning rate $\eta_\gamma$

$$\gamma_{t,k+1} = \gamma_{t,k} + \eta_\gamma \nabla_\gamma \log \int p(\mathcal{B}_{t+1}|\tilde{\mu}_t(\gamma_{t,k}) + \epsilon \circ \tilde{\sigma}_t(\gamma_{t,k}))\mathcal{N}(\epsilon; 0, I)d\epsilon, \tag{10}$$

The integral is evaluated by Monte-Carlo (MC) using $M$ samples $\epsilon_i \sim \mathcal{N}(\epsilon; 0, I)$, $i = 1, \ldots, M$

$$\int p(\mathcal{B}_{t+1}|\tilde{\mu}_t(\gamma_{t,k}) + \epsilon \circ \tilde{\sigma}_t(\gamma_{t,k}))\mathcal{N}(\epsilon; 0, I)d\epsilon \approx \frac{1}{M}\sum_{i=1}^M p(\mathcal{B}_{t+1}|\tilde{\mu}_t(\gamma_{t,k}) + \epsilon_i \circ \tilde{\sigma}_t(\gamma_{t,k})) \tag{11}$$

Inductive bias in the drift model is captured by $\gamma_t^0$, where $\gamma_{t,0} = 1$ encourages stationarity, while $\gamma_{t,0} = \gamma_{t-1,K}$ promotes temporal smoothness. In practice, we found $\gamma_{t,0} = 1$ was the most effective.

**Structure in the drift model.** The drift model can be defined to be shared across different *subsets* of parameters which reduces the expressivity of the drift model but also provides regularization to (10). We consider $\gamma_t$ to be either defined for each *parameter* or for each *layer*. See Section 5 for details as well as corresponding results in Appendix K.

**Interpretation of $\gamma_t$.** By linearising $\log p(\mathcal{B}_{t+1}|\theta)$ around $\mu_t$ and denoting $g_{t+1} = \nabla\widehat{\mathcal{L}}_{t+1}(\mu_t)$, we compute (8) in a closed form and get the following loss for $\gamma_t$ (see Appendix F) optimizing (9)

$$\mathcal{F}(\gamma_t) = 0.5(\tilde{\sigma}_t^2(\gamma_t) \circ g_{t+1})^\top g_{t+1} - (\gamma_t \circ \mu_t + (1 - \gamma_t) \circ \mu_0)^\top g_{t+1}, \tag{12}$$

Adding the $\ell_2$ penalty $\frac{1}{2}\lambda||\gamma_t - \gamma_t^0||^2$ encoding the starting point $\gamma_t^0$, gives us the closed form for $\gamma_t$

$$\gamma_t = \frac{\bar{g}_{t+1}^T(\bar{\mu}_t - \bar{\mu}_0) + K\lambda\gamma_{t,0}}{(\bar{g}_{t+1}\circ(\bar{\sigma}_0^2 - \bar{\sigma}_t^2))^T \bar{g}_{t+1} + K\lambda}, \tag{13}$$

where we assumed that $\gamma_t$ is shared for a subset of $K$ parameters (see paragraph about structure in drift model) indexed by $J_K = (j_1, \ldots, j_K)$ and $\bar{x} = (x_{j_1}, \ldots, x_{j_K})$ denotes a vector defined on this subset. We also clip parameters $\gamma_t$ to $[0, 1]$. The expression (13) gives us the geometric interpretation for $\gamma_t$. The value of $\gamma_t$ depends on the angle between $(\bar{\mu}_t - \bar{\mu}_0)$ and $\bar{g}_{t+1}$ When these vectors are aligned, $\gamma_t$ is high and is low otherwise. When these vectors are orthogonal or the gradient $\bar{g}_{t+1} \approx 0$, the value of $\gamma_t$ is heavily influenced by $\gamma_t^0$. Moreover, when $\bar{g}_{t+1} \approx 0$, we can interpret it as being close to a local minimum, i.e., stationary, which means that we want $\gamma_t \approx 1$, therefore adding the $\ell_2$ penalty is important. Also, when the norm of the gradients $\bar{g}_{t+1}$ is high, the value of $\gamma_t$ is encouraged to decrease, introducing the drift. This means that using $\gamma_t$ in the parameter update (see Section 3.5) encourages the norm of the gradient to stay small. In practice, we found that update (13) was unstable suggesting that linearization of the log-likelihood might not be a good approximation for learning $\gamma_t$.

### 3.4 Approximate Bayesian update of posterior $q_t(\theta)$ with BNNs

The optimization problem (6) for the per-parameter Gaussian $q(\theta) = \prod_j \mathcal{N}(\theta^j; \mu^j, \sigma^{j\,2})$ with a prior $q_t(\theta) = \prod_j \mathcal{N}(\theta^j; \mu_t^j, \sigma_t^{j\,2})$ can be written (see Appendix E.1) to minimize the following loss

$$\tilde{\mathcal{F}}_t(\mu, \sigma, \gamma_t) = \mathbb{E}_{\epsilon \sim \mathcal{N}(0;I)}\left[\widehat{\mathcal{L}}_{t+1}(\mu + \epsilon \circ \sigma)\right] + \sum_{j=1}^D \lambda_t^j \left[\frac{(\mu^j - \tilde{\mu}_t^j(\gamma_t))^2 + [\sigma^j]^2}{2[\tilde{\sigma}_t^j(\gamma_t)]^2} - \frac{1}{2}\log\left[\sigma^j\right]^2\right], \tag{14}$$

where $\lambda_t^j > 0$ are per-parameter temperature coefficients. The use of a small temperature parameter $\lambda > 0$ (shared for all NN parameters) was shown to improve the empirical performance of Bayesian Neural Networks [54]. Given that in (14), the variance $\tilde{\sigma}_t^{j\,2}(\gamma_t)$ can be small, in order to control the strength of the regularization, we propose to use the temperature per parameter $\lambda_t^j = \lambda \left[\sigma_t^j\right]^2$, where $\lambda > 0$ is a global constant. This leads to the following objective

$$\hat{\mathcal{F}}_t(\mu, \sigma, \gamma_t) = \mathbb{E}_{\epsilon \sim \mathcal{N}(0;I)}\left[\widehat{\mathcal{L}}_{t+1}(\mu + \epsilon \circ \sigma)\right] + \frac{\lambda}{2}\sum_j r_t^j \left[(\mu_j - \tilde{\mu}_t^j(\gamma_t))^2 + [\sigma^j]^2 - \left[\tilde{\sigma}_t^j(\gamma_t)\right]^2\log[\sigma^j]^2\right], \tag{15}$$

where the quantity $r_t^j = [\sigma_t^j]^2/[\sigma_t^j(\gamma_t)]^2$ is a relative change in the posterior variance due to the drift. The ratio $r_t^j = 1$ when $\gamma_t^j = 1$. For $\gamma_t^j < 1$ since typically $\sigma_t^{j\,2} < \sigma_0^{j\,2}$, the ratio is $r_t^j < 1$.

---

**Algorithm 1** *Soft-Reset* algoritm

---

**Input:** Data-stream $\mathcal{S}_T = \{\mathcal{B}_t\}_{t=1}^T$
Neural Network (NN) initializing distribution $p_{init}(\theta)$ and specific initialization $\theta_0 \sim p_{init}(\theta)$
Learning rate $\alpha_t$ for parameters and $\eta_\gamma$ for drift parameters
Number of gradient updates $K_\gamma$ on drift parameter $\gamma_t$
NN initial standard deviation (STD) scaling $\nu \leq 1$ (see (24)) and ratio $s$ defining $\frac{\sigma_t}{\nu \sigma_0}$.
**for** step $t = 0, 1, 2, \ldots, T$ **do**
    Receive batch of data $\mathcal{B}_{t+1}$
    Initialize drift parameters $\gamma_{t,0} = 1$
    **for** step $k = 0, 1, 2, \ldots, K_\gamma$ **do**
        Sample $\theta_0' \sim p_{init}(\theta)$
        Stochastic update (22) on drift parameter using specific initialization (26)
$$\gamma_{t,k+1} = \gamma_{t,k} + \eta_\gamma \nabla_\gamma \left[ \log p(\mathcal{B}_{t+1} | \gamma_t \theta_t + (1 - \gamma_t)\theta_0 + \theta_0' \circ \nu \sqrt{1 - \gamma_t^2 + \gamma_t^2 s^2}) \right]_{\gamma_t = \gamma_{t,k}}$$
    **end for**
    Get $\tilde{\theta}_t(\gamma_{t,K})$ with (18) and $\tilde{\alpha}_t(\gamma_{t,K})$ with (19)
    Update parameters $\theta_{t+1} = \tilde{\theta}_t(\gamma_{t,K}) - \tilde{\alpha}_t(\gamma_{t,K}) \circ \nabla_\theta \widehat{\mathcal{L}}_{t+1}(\tilde{\theta}_t(\gamma_{t,K}))$
**end for**

---

Thus, as long as the non stationarity is detected ($\gamma_t^j < 1$), the objective (15) favors the data term $\mathbb{E}_{\epsilon \sim \mathcal{N}(0;I)} \left[ \widehat{\mathcal{L}}_{t+1}(\mu + \epsilon \circ \sigma) \right]$ allowing the optimization to respond faster to changes in the data distribution. Let $\mu_{t+1,0} = \tilde{\mu}_t(\gamma_t)$ and $\sigma_{t+1,0} = \tilde{\sigma}_t(\gamma_t)$, and perform updates $K$ on (15)

$$\mu_{t+1,k+1} = \mu_{t+1,k} - \alpha_\mu \hat{\mathcal{F}}_t(\mu_{t+1,k}, \sigma_{t+1,k}, \gamma_t), \quad \sigma_{t+1,k+1} = \sigma_{t+1,k} - \alpha_\sigma \hat{\mathcal{F}}_t(\mu_{t+1,k}, \sigma_{t+1,k}, \gamma_t), \tag{16}$$

where $\alpha_\mu$ and $\alpha_\sigma$ are learning rates for the mean and for the standard deviation correspondingly. All derivations are provided in Appendix E.1. The full procedure is described in Algorithm 2.

## 3.5 Fast MAP update of posterior $q_t(\theta)$

As a faster alternative to propagating the posterior (6), we do MAP updates with the prior $p_0(\theta) = \prod_j \mathcal{N}(\theta^j; \mu_0^j; \sigma_0^{j2})$ and the approximate posterior $q_t(\theta) = \prod_j \mathcal{N}(\theta^j; \theta_t^j; \sigma_t^{j2} = s^2 \sigma_0^{j2})$, where $s \leq 1$ is a hyperparameter controlling the variance of $q_t(\theta)$. Since a fixed $s$ may not capture the true parameters variance, using a Bayesian method (see Section 3.4) is preferred but comes at a high computational cost (see Appendix I for discussion). The MAP update is given by (see Appendix E.2 for derivations) finding a minimum of the following proximal objective

$$\widehat{G}_{t+1}(\theta) = \widehat{\mathcal{L}}_{t+1}(\theta) + \frac{1}{2} \sum_{j=1}^D \frac{|\theta^j - \tilde{\theta}_t^j(\gamma_t)|^2}{\tilde{\alpha}_t^j(\gamma_t)} \tag{17}$$

where the regularization target for the parameter dimension $i$ is given by

$$\tilde{\theta}_t^j(\gamma_t) = \gamma_t^j \theta_t^j + (1 - \gamma_t^j)\mu_0^j \tag{18}$$

and the per-parameter learning rate is given as (assuming that $\alpha_t$ the base SGD learning rate)

$$\tilde{\alpha}_t^j(\gamma_t) = \alpha_t \left( (\gamma_t^j)^2 + \frac{1 - (\gamma_t^j)^2}{s^2} \right). \tag{19}$$

Linearising $\widehat{\mathcal{L}}_{t+1}(\theta)$ around $\tilde{\theta}_t(\gamma_t)$ and optimizing (17) for $\theta$ leads to (see Appendix E.2)

$$\theta_{t+1} = \tilde{\theta}_t(\gamma_t) - \tilde{\alpha}_t(\gamma_t) \circ \nabla_\theta \widehat{\mathcal{L}}_{t+1}(\tilde{\theta}_t(\gamma_t)), \tag{20}$$

For $\gamma_t^j = 1$, we recover the ordinary SGD update, while the values $\gamma_t^j < 1$ move the starting point of the modified SGD closer to the initialization as well as increase the learning rate. In Appendix D we describe additional practical choices made for the *Soft Resets* algorithm. Algorithm 1 describes the full procedure. Similarly to the Bayesian approach (16), we can do multiple updates on (17). This *Soft Resets Proximal* algorithm is given in Appendix E.2. Algorithm 3 describes the full procedure.

# 4 Related Work

**Plasticity loss in Neural Networks.** Our model shares similarities with reset-based approaches such as Shrink & Perturb (S&P) [2] and L2-Init [33]; however, whereas we learn drift parameters from data, these methods do not, leaving them vulnerable to mismatch between assumed non-stationarity and the actual realized non-stationarity in the data. Continual Backprop [13] or ReDO [47] apply resets in a data-dependent fashion, e.g. either based on utility or whether units are dead. But they use hard resets, and cannot amortize the cost of removing entire features. Interpretation (13) of $\gamma_t$ connects to the notion of parameters utility from [14], but this quantity is used to prevent catastrophic forgetting by decreasing learning rate for high $\gamma_t$. Our method increases the learning rate for low $\gamma_t$ to maximize adaptability, and is not designed to prevent catastrophic forgetting.

**Non-stationarity.** Non-stationarity arises naturally in a variety of contexts, the most obvious being continual and reinforcement learning. The structure of non-stationarity may vary from problem to problem. At one extreme, we have a *piece-wise stationary* setting, for example a change in the location of a camera generating a stream of images, or a hard update to the learner's target network in value-based deep RL algorithms. This setting has been studied extensively due to its propensity to induce *catastrophic forgetting* [e.g. 31, 45, 51, 10] and *plasticity loss* [13, 39, 38, 34]. At the other extreme, we can consider more gradual changes, for example due to improvements in the policy of an RL agent [40, 46, 42, 13] or shifts in the data generating process [36, 55, 20, 53]. Further, these scenarios might be combined, for example in *continual reinforcement learning* [31, 1, 13] where the reward function or transition dynamics could change over time.

**Non-stationary online convex optimization.** Non-stationary prediction has a long history in online convex optimization, where several algorithms have been developed to adapt to changing data [see, e.g., 25, 8, 22, 17, 21, 18, 29]. Our approach takes an inspiration from these works by employing a drift model as, e.g., [25, 21] and by changing learning rate as [29, 52]. Further, our OU drift model bears many similarities to the implicit drift model introduced in the update rule of [25] (see also [8, 17]), where the predictive distribution is mixed with a uniform distribution to ensure the prediction could change quickly enough if the data changes significantly, where in our case $p_0$ plays the same role as the uniform distribution.

**Bayesian approaches to non-stationary learning.** A standard approach is Variational Continual Learning [41], which focuses on preventing catastrophic forgetting and is an online version of "Bayes By Backprop" [5]. This method does not incorporate dynamical parameter drift components. In [35], the authors applied variational inference (VI) on non-stationary data, using the OU-process and Bayesian forgetting, but unlike in our approach, their drift parameter is not learned. Further, in [49], the authors considered an OU parameter drift model similar to ours, with an adaptable drift scalar $\gamma$ and analytic Kalman filter updates, but is applied over the final layer weights only, while the remaining weights of the network were estimated by online SGD. In [28], the authors propose to deal with non-stationarity by assuming that each parameter is a finite sum of random variables following different OU process. They derive VI updates on the posterior of these variables. Compared to this work, we learn drift parameters for every NN parameter rather than assuming a finite set of drift parameters. A different line of research assumes that the drift model is known and use different techniques to estimate the hidden state (the parameters) from the data: in [9], the authors use Extended Kalman Filter to estimate state and in [3], they propagate the MAP estimate of the hidden state distribution with $K$ gradient updates on a proximal objective similar to (47), whereas in Bayesian Online Natural Gradient (BONG) [27], the authors use natural gradient for the variational parameters.

# 5 Experiments

**Soft reset methods.** There are multiple variations of our method. We call the method implemented by Algorithm 1 with 1 gradient update on the drift parameter *Soft Reset*, while other versions show different parameter choices: *Soft Reset* ($K_\gamma = 10$) is a version with 10 updates on the drift parameter, while *Soft Reset* ($K_\gamma = 10$, $K_\theta = 10$) is the method of Algorithm 3 in Appendix E.2 with 10 updates on drift parameter, followed by 10 updates on NN parameters. *Bayesian Soft Reset* ($K_\gamma = 10$, $K_\theta = 10$) is a method implemented by Algorithm 2 with 10 updates on drift parameter followed by 10 updates on the mean $\mu_t$ and the variance $\sigma_t^2$ (uncertainty) for each NN parameter. Bayesian method performed the best overall but required higher computational complexity (see Appendix I). Unless specified, $\gamma_t$ is shared for all the parameters in each layer (separately for weight and biases).

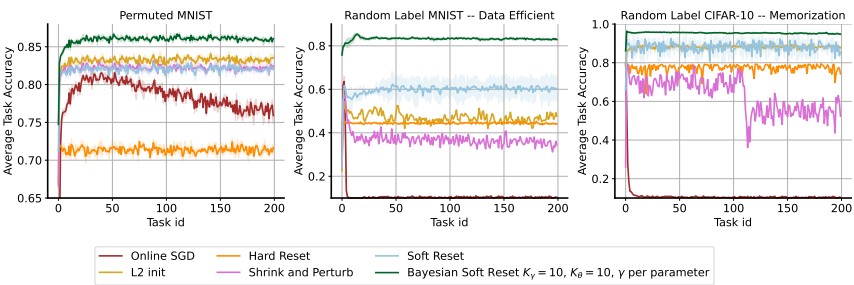

Figure 2: Plasticity benchmarks. **Left:** performance on *permuted MNIST*. **Center:** performance on *random-label MNIST* (data efficient). **Right:** performance on *random-label CIFAR-10* (memorization). The x-axis is the task id and the y-axis is the per-task training accuracy (52).

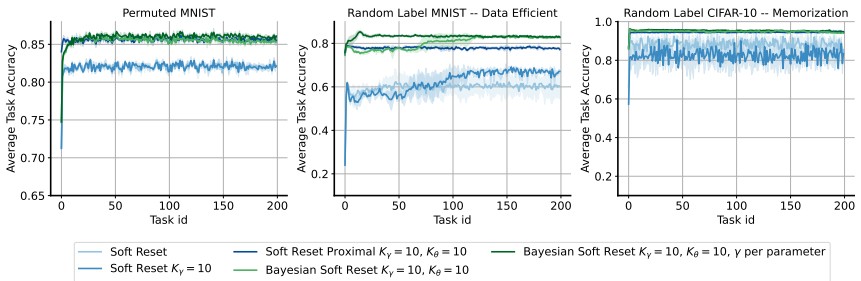

Figure 3: Different variants of *Soft Resets*. **Left:** performance on *permuted MNIST*. **Center:** performance on *random-label MNIST* (data efficient). **Right:** performance on *random-label CIFAR-10* (memorization). The x-axis is the task id and the y-axis is the per-task training accuracy (52).

**Loss of plasticity.** We analyze the performance of our method on *plasticity benchmarks* [34, 39, 38]. Here, we have a sequence of tasks, where each task consists of a fixed (for all tasks) subset of 10000 images images from either CIFAR-10 [32] or MNIST, where either pixels are permuted or the label for each image is randomly chosen. Several papers [34, 39, 38] study a *memorization* random-label setting where *SGD* can perfectly learn each task from scratch. To highlight the data-efficiency of our approach, we study the *data-efficient* setting where *SGD* achieves only 50% accuracy on each task when trained from scratch. Here, we expect that algorithms taking into account similarity in the data, to perform better. To study the impact of the non-stationarity of the input data, we consider *permuted MNIST* where pixels are randomly permuted within each task (the same task as considered by 34). As baselines, we use *Online SGD* and *Hard Reset* at task boundaries. We also consider *L2 init* [34], which adds $L2$ penalty $||\theta - \theta_0||^2$ to the fixed initialization $\theta_0$ as well as *Shrink&Perturb* [2], which multiplies each parameter by a scalar $\lambda \leq 1$ and adds random Gaussian noise with fixed variance $\sigma$. See Appendix H.1 for all details. As metrics, we use *average per-task online accuracy* (52), which is

$$\mathcal{A}_t = \frac{1}{N} \sum_{i=1}^{N} a_i^t,$$

where $a_i^t$ are the online accuracies collected on the task $t$ via $N$ timesteps, corresponding to the duration of the task. In Figure 5, we also use average accuracy over all $T$ tasks, i.e.

$$\mathcal{A}_T = \frac{1}{T} \sum_{t=1}^{T} \mathcal{A}_t$$

The results are provided in Figure 2. We observe that *Soft Reset* is always better than *Hard Reset* and most baselines despite the lack of knowledge of task boundaries. The gap is larger in the *data efficient* regime. Moreover, we see that *L2 Init* only performs well in the *memorization* regime, and achieves comparable performance to *Hard Reset* in the *data efficient* one. The method *L2 Init* could be viewed as an instantiation of our *Soft Reset Proximal* method optimizing (17) with $\gamma_t = 0$ at every step, which is sub-optimal when there is similarity in the data. *Bayesian Soft Reset* demonstrates significantly better performance overall, see also discussion below.

In Figure 3, we compare different variants of *Soft Reset*. We observe that adding more compute for estimating $\gamma_t$ (thus, estimating non-stationarity, $K_\gamma = 10$) as well as doing more updates on NN

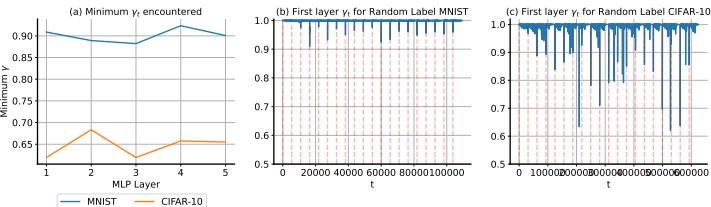

Figure 4: **Left:** the minimum encountered $\gamma_t$ for each layer on random-label MNIST and CIFAR-10. **Center:** the dynamics of $\gamma_t$ on the first 20 tasks on MNIST. **Right:** the same on CIFAR-10.

parameters (thus, more accurately adapting to non-staionarity, $K_\theta = 10$) leads to better performance. All variants of *Soft Reset* $\gamma_t$ parameters are shared for each NN layer, except for the Bayesian method. This variant is able to take advantage of a more complex *per-parameter* drift model, while other variants performed considerably worse, see Appendix K.4. We hypothesize this is due to the NN parameters uncertainty estimates $\sigma_t$ which Bayesian method provide, while others do not, which leads to a more accurate drift model estimation, since uncertainty is used in this update (10). But, this approach comes at a higher computational cost, see Appendix I. In Appendix K, we provide ablations of the structure of the drift model, as well as of the impact of learning the drift parameter.

**Qualitative behavior of *Soft Resets*.** For *Soft Reset*, we track the values of $\gamma_t$ for the first MLP layer when trained on random-label tasks studied above (only 20 tasks), as well as the minimum encountered value of $\gamma_t$ for each layer, which highlights the maximum amount of resets. Figure 4b,c shows $\gamma_t$ as a function of $t$, and suggests that $\gamma_t$ aggressively decreases at task boundaries (red dashed lines). The range of values of $\gamma_t$ depends on the task and on the layer, see Figure 4a. Overall, $\gamma_t$ changes more aggressively for long duration (memorization) random-label CIFAR-10 and less for shorter (data-efficient) random-label MNIST. See Appendix K.2 for more detailed results.

To study the behavior of *Soft Reset* under input distribution non-stationarity, we consider a variant of Permuted MNIST where each image is partitioned into patches of a given size. The non-stationarity is controlled by permuting the patches (not pixels). Figure 5a shows the minimum encountered $\gamma_t$ for each layer for different patch sizes. As the patch size increases and the problem becomes more stationary, the range of values for $\gamma_t$ is less aggressive. See Appendix K.3 for more detailed results.

**Impact of non-stationarity.** We consider a variant of random-label MNIST where for each task, an image has either a random or a true label. The label assignment is kept fixed throughout the task and is changed at task boundaries. We consider cases of $20\%$, $40\%$ and $60\%$ of random labels and we control the duration of each task (number of epochs). In total, the stream contains 200 tasks. In Figure 5b, we show performance of *Online SGD*, *Hard Reset* and in Figure 5c, the one of *Soft Reset* and of *Bayesian Soft Reset*. See Appendix H.2 for more details. The results suggest that for the shortest duration of the tasks, the performance of all the methods is similar. As we increase the duration of each of the task (moving along the x-axis), we see that both *Soft Resets* variants perform better than SGD and the gap widens as the duration increases. This implies that *Soft Resets* is more effective with infrequent data distribution changes. We also observe that Bayesian method performs better in all the cases, highlighting the importance of estimating uncertainty for NN parameters.

## 5.1 Reinforcement learning

**Reinforcement learning experiments.** We conduct Reinforcement Learning (RL) experiments in the highly off-policy regime, similarly to [43], since in this setting *loss of plasticity* was observed. We ran *SAC* [19] agent with default parameters from Brax [15] on the *Hopper*-v5 and *Humanoid*-v4 GYM [6] environments (from Brax [15]). To reproduce the setting from [43], we control the off-policyness of the agent by setting the *off-policy ratio* $M$ such that for every 128 environment steps, we do $128M$ gradient steps with batch size of 256 on the replay buffer. As baselines we consider ordinary *SAC*, hard-coded *Hard Reset* where we reset all the parameters $K = 5$ times throughout training (every 200000 steps), while keeping the replay buffer fixed (similarly to [43]). We employ our *Soft Reset* method as follows. After we have collected fresh data from the environment, we do one gradient update on $\gamma_t$ (shared for all the parameters within each layer) with batch size of 128 on this new chunk of data and the previously collected one, i.e., two chunks of data in total. Then we initialize $\tilde{\theta}_t(\gamma_t)$ and we employ the update rule (47) where the regularization $\tilde{\theta}_t(\gamma_t)$ is kept constant for all the off-policy gradient updates on the replay buffer. See Appendix H.3 for more details.

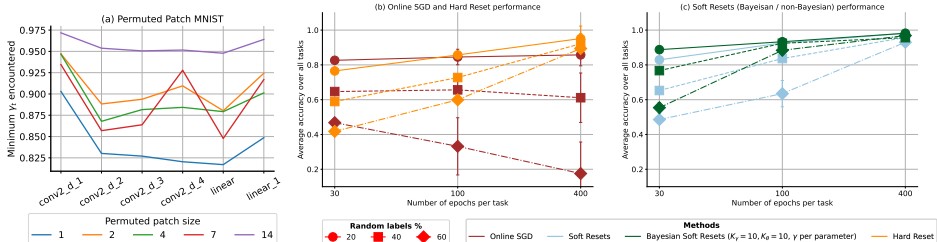

Figure 5: **(a)** the x-axis denotes the layer, the y-axis denotes the minimum encountered $\gamma_t$ for each convolutional and fully-connected layer when trained on permuted Patches MNIST, color is the patch size. The impact of non-stationarity on performance on random-label MNIST of Online SGD and Hard Reset is shown in **(b)** while the one of *Soft Resets* is shown in **(c)**. The x-axis denotes the number of epochs each task lasts, while the marker and line styles denote the percentage of random labels within each task, circle (solid) represents $20\%$, rectangle(dashed) $40\%$, while rhombus (dashed and dot) $60\%$. The y-axis denotes the average performance (over 3 seeds) on the stream of 200 tasks.

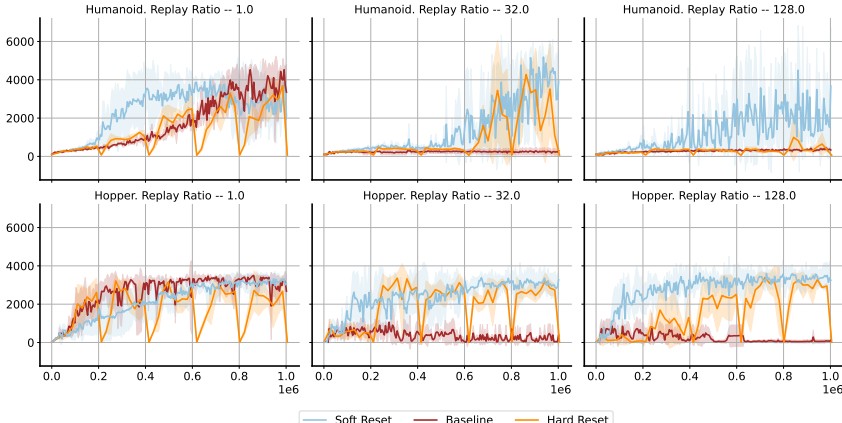

Figure 6: RL results. First row is humanoid, second is hopper. Each column corresponds to different replay ratio. The x-axis is the number of total timesteps, the y-axis the average reward. The shaded area denotes the standard deviation across 3 random seeds and the solid line indicates the mean.

The results are given in Figure 6. As the off-policy ratio increases, *Soft Reset* becomes more efficient than the baselines. This is consistent with our finding in Figure 5b,c, where we showed that the performance of *Soft Reset* is better when the data distribution is not changing fast. Figure 8 in Appendix H.3 shows the value of learned $\gamma_t$. It shows $\gamma_t$ mostly change for the value function and not for the policy indicating that the main source of non-stationarity comes from the value function.

## 6 Conclusion

Learning efficiently on non-stationary distributions is critical to a number of applications of deep neural networks, most prominently in reinforcement learning. In this paper, we have proposed a new method, *Soft Resets*, which improves the robustness of stochastic gradient descent to nonstationarities in the data-generating distribution by modeling the drift in Neural Network (NN) parameters. The proposed drift model implements *soft reset* mechanism where the amount of reset is controlled by the drift parameter $\gamma_t$. We showed that we could learn this drift parameter from the data and therefore we could learn *when* and *how far* to reset each Neural Network parameter. We incorporate the drift model in the learning algorithm which improves learning in scenarios with plasticity loss. The variant of our method which models uncertainty in the parameters achieves the best performance on plasticity benchmarks so far, highlighting the promise of the Bayesian approach. Furthermore, we found that our approach is particularly effective either on data distributions with a lot of similarity or on slowly changing distributions. Our findings open the door to a variety of exciting directions for future work, such as investigating the connection to continual learning and deepening our theoretical analysis of the proposed approach.

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

# A  Organization of the appendix

The appendix is organized as follows. In Appendix B we describe the Ornstein-Uhlenbeck process and connect it to the drift model (5), we use in the paper. In Appendix C, we discuss other potential choices for a drift model. In Appendix D, we describe additional practical tricks used for learning the drift model parameters. In Appendix E, we describe full Bayesian and maximum a-posteriori (MAP) algorithms to learn both drift parameters and NN parameters. In Appendix F, we provide the proof for the closed form solution derivation (12). In Appendix G, we briefly review the proximal SGD. In Appendix H, we provide experimental details. In Appendix I, we describe computational complexity for all the methods. In Appendix J, we provide sensitivity analysis for different parameters. In Appendix K, we provide additional ablations and experiments showing qualitative behavior of soft resets. In Appendix L, we show a toy example for the failure of SGD in the non-stationary regime. Finally, in Appendix M, we show how the proximal objective can be generalized to use any Gaussian drift model.

# B  Ornstein-Uhlenbeck process

We make use that the Ornstein-Uhlenbeck process [50] defines a SDE that can be solved explicitly and written as a time-continuous Gaussian Markov process with transition density between on $\mathbb{R}^D$

$$p(x_t|x_s) = \mathcal{N}(x_t; x_s e^{-(t-s)}, (1 - e^{-2(t-s)})\sigma_0^2 I),$$

for any pair of times $t > s$. Based on this as a drift model for the parameters $\theta_t$ (so $\theta_t$ is the state $x_t$) we use the conditional density

$$p(\theta_{t+1}|\theta_t, \gamma_t) = \prod_j \mathcal{N}(\theta_{t+1}^j; \theta_t^j \gamma_t^j, (1 - \gamma_t^{j\,2})\sigma_0^{j\,2}),$$

where $\gamma_t^j = e^{-\delta_t^j}$ and $\delta_t^j \geq 0$ corresponds to the learnable discretization time step. In other words, by learning $\gamma_t^j$ online we equivalently learn the amount of a continuous "time shift" $\delta_t^j$ between two consecutive states in the OU process. This essentially models parameter drift since e.g. if $\gamma_t^j = 1$, then $\delta_t^j = 0$ and there is no "time shift" which means that the next state/parameter remains the same as the previous one, i.e. $\theta_{t+1}^j = \theta_t^j$.

# C  Other choices of drift model

In this section, we discuss alternative choices of a drift model instead of (5).

**Independent mean and variance of the drift.**  We consider the drift model where the mean and the variance are not connected, i.e.,

$$p(\theta_{t+1}|\theta_t, \gamma_t, \beta_t) = \prod_j \mathcal{N}(\theta_{t+1}^j, \gamma_t^j \theta_t^j + (1 - \gamma_t^j)\mu_0^j; \beta_t^{j\,2}), \tag{21}$$

where parameters $\gamma_t^j \in [0, 1]$ control the mean of the distribution and $\beta_t^j > 0$ control the variance. When parameters $\beta_t^j$ are fixed to a constant, this would be similar to our experiment in Figure 16 where we assume known task boundaries and we do not estimate the drift parameters but assume it as a hyperparameter. Figure 16, left corresponds to the case when $\beta_t^j$ are fixed parameters independent from $\gamma_t^j$ whereas Figure 16, right corresponds to the case when $\beta_t^j = \sqrt{1 - \gamma_t^{j\,2}}\sigma_0^j$, i.e., when we use the drift model (5). We see from the results, using drift model (5) leads to a better performance. In case when $\beta_t^j$ are learned, estimating the parameters of this model will likely overfit to the noise since there is a lot of degrees of freedom.

**Shrink & Perturb [2].**  When we do not use the mean of the initialization, we can use the following drift model

$$p(\theta_{t+1}|\theta_t, \lambda_t, \beta_t) = \prod_j \mathcal{N}(\theta_{t+1}^j; \lambda_t^j \theta_t^j, \beta_t^{j\,2}),$$

for $\lambda_t^j \in [0, 1]$ and $\beta_t^j > 0$. Similarly to the case of (21), estimating both parameters $\lambda_t^j$ and $\beta_t^j$ from the data will likely overfit to the noise.

**Arbitrary linear model.** We can use the arbitrary linear model of the form

$$p(\theta_{t+1}|\theta_t, A_t, B_t) = \mathcal{N}(\theta_{t+1}; A_t\theta_t, B_t),$$

but estimating the parameters $A_t$ and $B_t$ has too many degrees of freedom and will certainly overfit.

**Gaussian Spike & Slab** We consider a Gaussian [11] approximation to Spike & Slab [26] prior

$$p(\theta_{t+1}|\theta_t, \gamma_t) = \prod_j \left[ \gamma_t^j p(\theta_{t+1}^j|\theta_t^j) + (1 - \gamma_t^j)p_0^j(\theta_{t+1}^j) \right],$$

with binary variables $\gamma_t^j \in \{0, 1\}$, which is a product of mixtures of Gaussian distributions $p(\theta_{t+1}^j|\theta_t^j) = \mathcal{N}(\theta_{t+1}^j; \theta_t^j, \sigma^{j2})$ centred around the previous parameter $\theta_t^j$ and initializing distributions $p_0(\theta_{t+1}^j) = \mathcal{N}(\theta_{t+1}^j; \mu_0^j, \sigma_0^{j2})$. This model, however, implements the mechanism of Hard reset as opposed to the soft ones. Moreover, estimating such a model and incorporating it into a learning update is more challenging since the mixture of Gaussian is not conjugate with respect to a Gaussian which will make the KL term (34) to be computed only approximately via Monte Carlo updates.

# D    Practical implementations of the drift model estimation

**Stochastic approximation for drift parameters estimation** In practice, we use $M = 1$, which leads to the stochastic approximation

$$\int p(\mathcal{B}_{t+1}|\tilde{\mu}_t(\gamma_t^k) + \epsilon \circ \sigma_t(\gamma_t^k))\mathcal{N}(\epsilon; 0, I)d\epsilon \approx p\left(\mathcal{B}_{t+1}|\tilde{\mu}_t(\gamma_t^k) + \epsilon \circ \sigma_t(\gamma_t^k)\right) \tag{22}$$

**Using NN initializing distribution.** In the drift model (5), we assume that the initial distribution over parameters is given by $p_0(\theta) = \prod_j \mathcal{N}(\theta^j; \mu_0^j, \sigma_0^{j2})$. In practice, we have access to the NN initializer $p_{init}(\theta) = \prod_j \mathcal{N}(\theta^j; 0, \sigma_0^{j2})$ where $\mu_0^j = 0$ (for most of the NNs). This means that we can replace $\epsilon$ from (10) by $\frac{1}{\sigma_0}\theta_0'$ where $\theta_0' \sim p_{init}(\theta)$. This means that the term in (22) can be replaced by

$$p\left(\mathcal{B}_{t+1}|\tilde{\mu}_t(\gamma_t^k) + \epsilon \circ \sigma_t(\gamma_t^k)\right) = p\left(\mathcal{B}_{t+1}|\tilde{\mu}_t(\gamma_t^k) + \theta_0' \circ \sqrt{1 - \gamma_t^2 + \gamma_t^2\frac{\sigma_t^2}{\sigma_0^2}}\right), \tag{23}$$

where we used the fact that $\sigma_t^2(\gamma_t) = \gamma_t^2\sigma_t^2 + (1 - \gamma_t^2)\sigma_0^2$. Note that in (23), we only need to know the ratio $\frac{\sigma_t^2}{\sigma_0^2}$ rather than both of these. We will see that in Section 3.5, only this ratio is used for the underlying algorithm. Finally, in practice, we can tie $p_0(\theta)$ to the *specific* initialization $\theta_0 \sim p_{init}(\theta)$. It was observed emprically [34] that using a specific initialization in gradient updates led to better performance than using samples from the initial distribution. This would imply that

$$p_0(\theta) = \prod_j \mathcal{N}(\theta^j; \theta_0^j, \tilde{\sigma}_0^{j2}), \tag{24}$$

with $\tilde{\sigma}_0^{j2} = \nu^2\sigma_0^{j2}$. The parameter $\nu \leq 1$ accounts for the fact that the distribution $p_0(\theta)$ should have lower than $p_{init}(\theta)$ variance since it uses the specific initializaiton from the prior. This modification would imply the following modification on the drift model term (23)

$$p\left(\mathcal{B}_{t+1}|\mu_t(\gamma_t^k) + \epsilon \circ \sigma_t(\gamma_t^k)\right) = p\left(\mathcal{B}_{t+1}|\mu_t(\gamma_t^k) + \theta_0' \circ \nu\sqrt{1 - \gamma_t^2 + \gamma_t^2\frac{\sigma_t^2}{\nu^2\sigma_0^2}}\right) \tag{25}$$

If we denote $s = \frac{\sigma_t}{\nu\sigma_0}$, then we can re-write (25) to be

$$p\left(\mathcal{B}_{t+1}|\mu_t(\gamma_t^k) + \epsilon \circ \sigma_t(\gamma_t^k)\right) = p\left(\mathcal{B}_{t+1}|\mu_t(\gamma_t^k) + \theta_0' \circ \nu\sqrt{1 - \gamma_t^2 + \gamma_t^2 s^2}\right) \tag{26}$$

In practice, we can treat $s$ to be a constant hyperparameter.

# E Learning parameters with estimated drift models

In this section, we provide a Bayesian Neural Network algorithm to learn the distributions of NN parameters when there is a drift in the data distribution. Moreover, we provide a Maximum a-Posteriori (MAP) like inference algorithm which does not require to learn the distributions over parameters, but simply propagates the MAP estimate over these.

## E.1 Bayesian Neural Networks algorithm

In this section, we describe an algorithm for parameters update based on Bayesian Neural Networks (BNN). It is based on the online variational Bayes setting described below.

Let the family of distributions over parameters be

$$\mathcal{Q} = \{q(\theta) : q(\theta) \sim \prod_{j=1}^{D} \mathcal{N}(\theta^j; \mu^j, \sigma^{j\,2}); \theta = (\theta^1, \dots, \theta^D)\}, \tag{27}$$

which is the family of Gaussian mean-field distributions over parameters $\theta \in \mathbb{R}^D$. Let $\Gamma_t = (\gamma_1, \dots, \gamma_t)$ be the history of observed parameters of the drift model and $\mathcal{S}_t = \{\mathcal{B}_1, \dots, \mathcal{B}_t\}$ be the history of observed data. We denote by $q_t(\theta) \triangleq q_t(\theta|\mathcal{S}_t, \Gamma_{t-1}) \in \mathcal{Q}$ the Gaussian *approximate posterior* at time $t$ with mean $\mu_t$ and variance $\sigma_t^2$ for every parameter. The approximate predictive look-ahead prior is given by

$$q_t(\theta|\gamma_t) = \int q_t(\theta_t) p(\theta|\theta_t, \gamma_t) d\theta_t = \mathcal{N}(\theta; \tilde{\mu}_t(\gamma_t), \tilde{\sigma}_t^2(\gamma_t)), \tag{28}$$

that has parameters

$$\tilde{\mu}_t^j(\gamma_t) = \gamma_t^j \mu_t^j + (1 - \gamma_t^j)\mu_0^j$$
$$\tilde{\sigma}_t^{j\,2}(\gamma_t) = \gamma_t^2 \sigma_t^{j\,2} + (1 - \gamma_t^{j\,2})\sigma_0^{j\,2} \tag{29}$$

To see this, we will use the law of total expectation and the law total variance. For two random variables $X$ and $Y$ defined on the same space, law of total expectation says

$$\mathbb{E}[Y] = \mathbb{E}[\mathbb{E}[Y|X]]$$

and the law of total variance says

$$\mathbb{V}[Y] = \mathbb{E}[\mathbb{V}[Y|X]] + \mathbb{V}[\mathbb{E}[Y|X]]$$

In our case, from the drift model (5), we have the conditional distribution

$$\theta^j|\theta_t^j = \gamma_t^j \theta_t^j + (1 - \gamma_t^j)\mu_0^j + \epsilon^j \sqrt{(1 - \gamma_t^{j\,2})\sigma_0^{j\,2}}, \epsilon^j \sim \mathcal{N}(0; 1) \tag{30}$$

From (30), we have

$$\mathbb{E}[\theta^j|\theta_t^j] = \gamma_t^j \theta_t^j + (1 - \gamma_t^j)\mu_0^j$$
$$\mathbb{V}[\theta^j|\theta_t^j] = (1 - \gamma_t^{j\,2})\sigma_0^{j\,2}$$

From here, we have that the mean is given by

$$\mathbb{E}[\theta] = \mathbb{E}[\mathbb{E}[\theta|\theta_t]] = \gamma_t^j \mu_t^j + (1 - \gamma_t^j)\mu_0^j \tag{31}$$

and the variance is given by

$$\mathbb{V}[\theta] = \mathbb{E}[\mathbb{V}[\theta|\theta_t]] + \mathbb{V}[\mathbb{E}[\theta|\theta_t]]$$
$$\mathbb{V}[\theta] = (1 - \gamma_t^{j\,2})\sigma_0^2 + \gamma_t^2 \theta_t^{j\,2} \tag{32}$$

Now, we note that $q_t(\theta|\gamma_t)$ is a Gaussian and its parameters are given by $\mathbb{E}[\theta^j] = \gamma_t^j \mu_t^j + (1 - \gamma_t^j)\mu_0^j$ from (31) and by $\mathbb{V}[\theta^j] = (1 - \gamma_t^{j\,2})\sigma_0^{j\,2} + \gamma_t^{j\,2}\theta_t^{j\,2}$ from (32). Then, for new data $\mathcal{B}_{t+1}$ at time $t + 1$, the *approximate predictive log-likelihood* equals to

$$\log q_t(\mathcal{B}_{t+1}|\gamma_t) = \log \int p(\mathcal{B}_{t+1}|\theta) q_t(\theta|\gamma_t) d\theta.$$

We are looking for a new approximate posterior $q_{t+1}(\theta)$ such that

$$q_{t+1}(\theta) = \arg \min_q \mathbb{KL}\left[q(\theta)||p(\mathcal{B}_{t+1}|\theta) q_t(\theta|\gamma_t)\right] \tag{33}$$

The optimization problem (33) can be written as minimization of the following loss

$$\mathcal{F}_t(\theta, \gamma_t) = \mathbb{E}_q\left[\widehat{\mathcal{L}}_{t+1}(\theta)\right] + \mathbb{KL}\left[q(\theta)||q_t(\theta|\gamma_t)\right], \tag{34}$$

Using the fact that we are looking for a member $q \in \mathcal{Q}$ from (27), we can write the objective (34) as

$$\mathcal{F}_t(\mu, \sigma, \gamma_t) = \mathbb{E}_{\epsilon \sim \mathcal{N}(0;I)}\left[\widehat{\mathcal{L}}_{t+1}(\mu + \epsilon \circ \sigma)\right] + \mathbb{KL}\left[q(\theta)||q_t(\theta|\gamma_t)\right],$$

where we used the reparameterisation trick for the loss term. We now expand the regularization term to get

$$\mathcal{F}_t(\mu, \sigma, \gamma_t) = \mathbb{E}_{\epsilon \sim \mathcal{N}(0;I)}\left[\widehat{\mathcal{L}}_{t+1}(\mu + \epsilon \circ \sigma)\right] + \sum_j\left[\frac{(\mu^j - \tilde{\mu}_t^j(\gamma_t))^2 + \sigma^{j2}}{2\tilde{\sigma}_t^{j2}(\gamma_t)} - \frac{1}{2}\log\sigma_j^2\right] \tag{35}$$

Since the posterior variance of NN parameters may become small, the optimization of (35) may become numerically unstable due to division by $\tilde{\sigma}_t^{j2}(\gamma_t)$. It was shown [54] that using small temperature on the prior led to better empirical results when using Bayesian Neural Networks, a phenomenon known as cold posterior. Here, we define a temperature per-parameter, i.e., $\lambda_t^j > 0$ for every time-step $t$, such that the objective above becomes

$$\tilde{\mathcal{F}}_t(\mu, \sigma, \gamma_t; \{\lambda_t^j\}_j) = \mathbb{E}_{\epsilon \sim \mathcal{N}(0;I)}\left[\widehat{\mathcal{L}}_{t+1}(\mu + \epsilon \circ \sigma)\right] + \sum_j \lambda_t^j\left[\frac{(\mu^j - \tilde{\mu}_t^j(\gamma_t))^2 + \sigma^{j2}}{2\tilde{\sigma}_t^{j2}(\gamma_t)} - \frac{1}{2}\log\sigma^{j2}\right]$$
$$\tag{36}$$

As said above, it is common to use the same temperature $\lambda_t^j = \lambda$ for all the parameters. In this work, we propose the specific choice of the temperature to be

$$\lambda_t^j = \lambda\sigma_t^{j2}, \tag{37}$$

where $\lambda > 0$ is some globally chosen temperature parameter. This leads to the following objective

$$\hat{\mathcal{F}}_t(\mu, \sigma, \gamma_t; \{r_t^j\}_j) = \mathbb{E}_{\epsilon \sim \mathcal{N}(0;I)}\left[\widehat{\mathcal{L}}_{t+1}(\mu + \epsilon \circ \sigma)\right] + \frac{1}{2}\sum_j r_t^j\left[(\mu_j - \tilde{\mu}_t^j(\gamma_t))^2 + \sigma^{j2} - \tilde{\sigma}_t^{j2}(\gamma_t)\log\sigma^{j2}\right],$$
$$\tag{38}$$

where the quantity $r_t^j$ is defined as

$$r_t^j = \frac{\sigma_t^{j2}}{\tilde{\sigma}_t^{j2}(\gamma_t)} = \frac{\sigma_t^{j2}}{\gamma_t^{j2}\sigma_t^{j2} + (1-\gamma_t^{j2})\sigma_0^{j2}}, \tag{39}$$

which represents the relative change in the posterior variance due to the drift. In the exact stationary case, when $\gamma_t^j = 1$, this ratio is $r_t^j = 1$ while for $\gamma_t^j < 1$, since typically $\sigma_t^{j2} < \sigma_0^{j2}$, we have $r_t^j < 1$. This means that in the non-stationary case, the strength of the regularization in (38) in favor of the data term $\mathbb{E}_{\epsilon \sim \mathcal{N}(0;I)}[\mathcal{L}_{t+1}(\mu + \epsilon \circ \sigma)]$, allowing the optimization to respond faster to the change in the data distribution. In practice, this data term is approximated via Monte-Carlo, i.e.

$$\mathbb{E}_{\epsilon \sim \mathcal{N}(0;I)}\left[\widehat{\mathcal{L}}_{t+1}(\mu + \epsilon\sigma)\right] \sim \frac{1}{M}\sum_{i=1}^M \widehat{\mathcal{L}}_{t+1}(\mu + \epsilon_i\sigma) \tag{40}$$

To find new parameters, $\mu_{t+1}$ and $\sigma_{t+1}$, we let $\mu_{t+1,0} = \tilde{\mu}_t(\gamma_t)$ and $\sigma_{t+1,0} = \tilde{\sigma}_t(\gamma_t)$ where corresponding quantities are defined in (29) and perform multiple $K$ updates on (38)

$$\mu_{t+1,k+1} = \mu_{t+1,k} - \alpha_\mu\hat{\mathcal{F}}_t(\mu_{t+1,k}, \sigma_{t+1,k}, \gamma_t, \{r_t^j\}_j),$$
$$\sigma_{t+1,k+1} = \sigma_{t+1,k} - \alpha_\sigma\hat{\mathcal{F}}_t(\mu_{t+1,k}, \sigma_{t+1,k}, \gamma_t, \{r_t^j\}_j),$$

where $\alpha_\mu$ and $\alpha_\sigma$ are corresponding learning rates.

**Practical considerations.** As a prior distribution, we use

$$p_0(\theta) = \prod_j \mathcal{N}(\theta^j; 0, \sigma_0^j), \tag{41}$$

with some $\sigma_0^j > 0$. In practice, we often have access to the NN library with pre-defined $\tilde{\sigma}_0^j$. We allow more flexbility and define

$$\sigma_0^j = \nu \tilde{\sigma}_0^j, \tag{42}$$

with some constant $\nu \in (0, 1]$, allowing to scale the NN library variance.

When using Bayesian Neural Networks (BNN), one question is to how to initialize the variational posterior $q_0(\theta)$. We found that using specific initialization $\theta_0 \sim p_{init}(\theta)$ as the mean of the variational posterior $q_0(\theta)$, where $p_{init}(\theta)$ is the NN initializing distribution, worked significantly better than using 0 (which is the mean of $p_0(\theta)$). Therefore, we propose to initialize $q_0(\theta)$ as

$$q_0(\theta) = \prod_j \mathcal{N}(\theta^j; \theta_0^j, \sigma_{init}^{j}{}^2),$$

where $\theta_0 \sim p_{init}(\theta)$ and $\sigma_{init}^{j}{}^2$ is defined as

$$\sigma_{init}^{j}{}^2 = \pi^2 \sigma_0^{j\,2}, \tag{43}$$

and $\sigma_0^j$ is the variance of the prior distribution $p_0(\theta)$, see (41). Here, $\pi \in (0, 1]$ is a constant allowing us to scale down the initial variance of $q_0(\theta)$ with respect to the prior variance $\sigma_0^j$.

The full algorithm of learning the drift parameters $\gamma_t$ as well as learning the Bayesian Neural Network parameters using the procedure above is given in Algorithm 2.

### E.2 Modified SGD with drift model

Instead of propagating the posterior (6), we do MAP updates on (4) with the prior $p_0(\theta) = \prod_j \mathcal{N}(\theta^j; \mu_0^j; \sigma_0^{j\,2})$ and the posterior $q_t(\theta) = \prod_j \mathcal{N}(\theta^j; \theta_t^j; s^2 \sigma_0^{j\,2})$, where $s \leq 1$ is hyperparameter controlling the variance $\sigma_t^2$ of the posterior $q_t(\theta)$. Since fixed $s$ may not capture the true parameters variance, using Bayesian method (see Appendix E.1) is preferred but comes at a high computational cost. Instead of Bayesian update (33), we consider maximum a-posteriori (MAP) update

$$\max_\theta \log p(\mathcal{B}_{t+1}|\theta) + \log q_t(\theta|\gamma_t),$$

with $q_t(\theta|\gamma_t)$ given by (28). Using the definition of $q_t(\theta|\gamma_t)$, we get the following problem

$$\max_\theta -\widehat{\mathcal{L}}_{t+1}(\theta) - \sum_j \lambda_t^j \left[ \frac{(\mu_i - \tilde{\mu}_t^j(\gamma_t))^2}{2\tilde{\sigma}_t^{j\,2}(\gamma_t)} \right], \tag{44}$$

where similarly to (36), we use a per-parameter temperature $\lambda_t^j \geq 0$. We choose temperature to be equal to

$$\lambda_t^j = s^2 \sigma_0^{j\,2} \lambda,$$

where $\lambda$ is some constant. Such choice of temperature is motivated by the same logic as in (37) – it is a constant multiplied by the posterior variance $\sigma_t^{j\,2} = s^2 \sigma_0^{j\,2}$. With such choice of temperature, maximizing (44) is equivalent to minimizing

$$\widehat{G}_{t+1}(\theta; \lambda) = \widehat{\mathcal{L}}_{t+1}(\theta) + \frac{\lambda}{2} \sum_{j=1}^D \frac{|\theta^j - \tilde{\theta}_t^j(\gamma_t)|^2}{r_t^j(\gamma)} \tag{45}$$

where the regularization target for the dimension $i$ is

$$\tilde{\theta}_t^j(\gamma_t) = \gamma_t^j \theta_t^j + (1 - \gamma_t^j) \mu_0^j \tag{46}$$

and the constant $r_t^j(\gamma)$ is given by

$$r_t^j(\gamma_t) = \left( (\gamma_t^j)^2 + \frac{1 - (\gamma_t^j)^2}{s^2} \right)$$

We can perform $K$ gradient updates (45) with a learning rate $\alpha_t$ starting from $\theta_{t+1}^0 = \tilde{\theta}_t(\gamma_t)$,

$$\theta_{t+1}^{k+1} = \theta_{t+1}^k - \alpha_t(\gamma_t) \circ \nabla_\theta \widehat{G}_{t+1}(\theta_{t+1}^k; \lambda), \tag{47}$$

where the vector-valued learning rate $\alpha_t(\gamma_t)$ is given by

$$\tilde{\alpha}_t(\gamma_t) = \alpha_t \tilde{r}_t^j(\gamma_t) = \alpha_t \left( (\gamma_t^j)^2 + \frac{1 - (\gamma_t^j)^2}{s^2} \right), \tag{48}$$

with $\alpha_t$ the base learning rate. Note that doing one update is equivalent to modified SGD method (20). Doing multiple updates on (47) allows us to perform multiple computations on the *same* data. The corresponding algorithm is given in Algorithm 3.

---

**Algorithm 2** Bayesian *Soft-Reset* algorithm

---

**Input:** Data-stream $\mathcal{S}_T = \{\mathcal{B}_t\}_{t=1}^T$
Global temperature parameter $\lambda \geq 0$.
Learning rate for the mean $\alpha_\mu$ and for the standard deviation $\alpha_\sigma$
Number of gradient updates $K_\theta$ to be applied on $\mu$ and $\sigma$
Number of Monte-Carlo samples $M_\theta$ for estimating $\mu$ and $\sigma$ in (40)
Number of gradient updates $K_\gamma$ on drift parameter $\gamma_t$ in (10)
Number of Monte-Carlo samples $M_\gamma$ to estimate $\gamma_t$ in (11)
Learning rate $\eta_\gamma$ for drift parameters
Initial drift parameters $\gamma_0 = 1$ for every iteration.
**Initialization parameters**:
Neural Network initial variance for every parameter $\tilde{\sigma}_0^{j\,2}$ coming from standard NN library
Initial prior variance rescaling $\nu \in (0, 1]$.
Prior variance $\sigma_0^{j\,2} = \nu^2 \tilde{\sigma}_0^{j\,2}$ given by (42)
Initial NN parameters $\theta_0 \sim p_{init}(\theta)$
Initial posterior variance rescaling $\pi \in (0, 1]$.
Initial posterior variance $\sigma_{init}^{j}{}^2 = \pi^2 \sigma_0^{j\,2}$ given by (43)
**Initialization**:
Initialize zero-mean prior distribution (41) as $p_0(\theta) = \prod_j \mathcal{N}(\theta^j; 0, \sigma_0^{j2})$
Initialize posterior $q_0(\theta) = \prod_j \mathcal{N}(\theta^j; \theta_0^j, \sigma_{init}^j{}^2)$
**for** step $t = 0, 1, 2, \ldots, T$ **do**
    Receive batch $\mathcal{B}_{t+1}$
    Current posterior is given by $q_t = \mathcal{N}(\theta; \mu_t, \sigma_t^2)$
    **Estimating the drift**
    Initialize drift parameter $\gamma_t^0 = \gamma_0$.
    Compute drifted prior mean $\tilde{\mu}_t^j(\gamma_t) = \gamma_t^j \mu_t^j$ assuming zero-mean prior $\mu_0^j = 0$
    Compute drifted prior variance $\tilde{\sigma}^{j2}(\gamma_t) = \gamma_t^{j\,2}\sigma_t^{j\,2} + (1 - \gamma_t^{j\,2})\sigma_0^{j\,2}$
    **for** $k = 0, \ldots, K_\gamma - 1$ **do**
        Sample Gaussian noise $\epsilon_i \sim \mathcal{N}(\epsilon_i; 0, I)$, $i = 1, \ldots, M_\gamma$
        $\gamma_t^{k+1} = \gamma_t^k + \eta_\gamma \nabla_\gamma \log \frac{1}{M_\gamma} \sum_{i=1}^{M_\gamma} p(\mathcal{B}_{t+1} | \tilde{\mu}_t(\gamma_t^k) + \epsilon_i \circ \tilde{\sigma}_t(\gamma_t^k))$
    **end for**
    **Updating variational posterior**
    Let $\mu_{t+1,0}^j = \gamma_t^j \mu_t^j$, $\sigma_{t+1,0}^j = \sqrt{\gamma_t^{j\,2}\sigma_t^{j\,2} + (1 - \gamma_t^{j\,2})\sigma_0^2}$, see (29)
    Let $r_t^j = \frac{\sigma_{t,i}^2}{\sigma_t^2(\gamma_t)} = \frac{\sigma_t^{j\,2}}{\gamma_t^2 \sigma_t^{j\,2} + (1 - \gamma_t^{j\,2})\sigma_0^{j\,2}}$ using (39)
    **for** $k = 0, \ldots, K_\theta - 1$ **do**
        Using the definition of $\hat{\mathcal{F}}_t$ from (38), do
        $\mu_{t+1,k+1} = \mu_{t+1,k} - \alpha_\mu \hat{\mathcal{F}}_t(\mu_{t+1,k}, \sigma_{t+1,k}, \gamma_t, \{r_t^j\}_j, \lambda)$
        $\sigma_{t+1,k+1} = \sigma_{t+1,k} - \alpha_\sigma \hat{\mathcal{F}}_t(\mu_{t+1,k}, \sigma_{t+1,k}, \gamma_t, \{r_t^j\}_j, \lambda)$
    **end for**
**end for**

---

## F  Proof of linearisation

**Interpretation of $\gamma_t$.**  By linearising $\log p(\mathcal{B}_{t+1}|\theta)$ around $\mu_t$, we can simplify (8) to get

$$\mathcal{F}(\gamma_t) = (\gamma_t \circ \mu_t + (1 - \gamma_t) \circ \mu_0)^T g_{t+1} - 0.5(\sigma_t^2(\gamma_t) \circ g_{t+1})^T g_{t+1} - \tfrac{1}{2}\lambda \sum_{j=1}^K (\gamma_t^j - \gamma_{t,0}^j)^2,$$

where $\circ$ denotes elementwise product, $g_t = \nabla \widehat{\mathcal{L}}_{t+1}(\mu_t)$ is the negative gradient of the loss (1) evaluated at $\mu_t$ and we added the $\ell_2$-penalty $\frac{1}{2}\lambda ||\gamma_t - \gamma_{t,0}||^2$ to take into account the initialization.

**Proof**. We assume that the following linearisation is correct

$$\log p(\mathcal{B}_{t+1}|\theta) \sim \log p(\mathcal{B}_{t+1}|\mu_t) + g_{t+1}^T(\theta - \mu_t),$$

where

$$g_{t+1} = -\nabla_\theta \log p(\mathcal{B}_{t+1}|\theta = \mu_t) = \nabla_\theta \widehat{\mathcal{L}}_{t+1}(\mu_t)$$

---

**Algorithm 3** Proximal *Soft-Reset* algoritm

---

**Input:** Data-stream $\mathcal{S}_T = \{\mathcal{B}_t\}_{t=1}^T$
Neural Network (NN) initializing distribution $p_{init}(\theta)$ and specific initialization $\theta_0 \sim p_{init}(\theta)$
Learning rate $\alpha_t$ for parameters and $\eta_\gamma$ for drift parameters
Number of gradient updates $K_\gamma$ on drift parameter $\gamma_t$
Number of gradient updates $K_\theta$ on NN parameters
Global temperature parameter $\lambda \geq 0$
NN initial standard deviation (STD) scaling $\nu \leq 1$ (see (24)) and ratio $s = \frac{\sigma_t}{\nu \sigma_0}$.
**for** step $t = 0, 1, 2, \ldots, T$ **do**
    Receive batch $\mathcal{B}_{t+1}$
    Initialize drift parameter $\gamma_t^0 = 1$
    **for** step $k = 0, 1, 2, \ldots, K_\gamma$ **do**
        Sample $\theta_0' \sim p_{init}(\theta)$
        Stochastic update (22) on drift parameter using specific initialization (25)
$$\gamma_t^{k+1} = \gamma_t^k + \eta_\gamma \nabla_\gamma \left[ \log p(\mathcal{B}_{t+1}|\gamma_t \theta_t + (1-\gamma_t)\theta_0 + \theta_0' \circ \nu \sqrt{1 - \gamma_t^2 + \gamma_t^2 s^2}) \right]_{\gamma_t = \gamma_t^k}$$
    **end for**
    Initialize $\theta_{t+1}^0 = \theta_t(\gamma_t^K)$ with (46) and use $\alpha_t(\gamma_t^K) = \alpha_t \left( (\gamma_t^i)^2 + \frac{1 - (\gamma_t^i)^2}{s^2} \right)$ with (48)
    **for** step $k = 0, 1, 2, \ldots, K_\theta$ **do**
        $\theta_{t+1}^{k+1} = \theta_{t+1}^k - \alpha_t(\gamma_t) \circ \nabla_\theta \widehat{G}_{t+1}(\theta_{t+1}^k; \lambda)$
    **end for**
**end for**

---

Then, we have
$$p(\mathcal{B}_{t+1}|\theta) \sim p(\mathcal{B}_{t+1}|\mu_t) \exp^{g_{t+1}^T(\theta - \mu_t)}$$
Let's write the integral from (8)
$$\log \int p(\mathcal{B}_{t+1}|\theta) \exp^{-\frac{1}{2}(\theta - \mu_t(\gamma_t))^T \Sigma_t^{-1}(\gamma_t)(\theta - \mu_t(\gamma_t))} d\theta \frac{1}{\sqrt{(2\pi)^D |\Sigma_t(\gamma_t)|}} =$$
$$\log \int p(\mathcal{B}_{t+1}|\mu_t) \exp^{g_{t+1}^T(\theta - \mu_t)} \exp^{-\frac{1}{2}(\theta - \mu_t(\gamma_t))^T \Sigma_t^{-1}(\gamma_t)(\theta - \mu_t(\gamma_t))} d\theta \frac{1}{\sqrt{(2\pi)^D |\Sigma_t(\gamma_t)|}} =$$
$$\log p(\mathcal{B}_{t+1}|\mu_t) + \log \int \exp^{g_{t+1}^T(\theta - \mu_t)} \exp^{-\frac{1}{2}(\theta - \mu_t(\gamma_t))^T \Sigma_t^{-1}(\gamma_t)(\theta - \mu_t(\gamma_t))} d\theta \frac{1}{\sqrt{(2\pi)^D |\Sigma_t(\gamma_t)|}}$$

Consider only the exp term inside the integral:
$$g_{t+1}^T(\theta - \mu_t) - \frac{1}{2}(\theta - \mu_t(\gamma_t))^T \Sigma_t^{-1}(\gamma_t)(\theta - \mu_t(\gamma_t)) =$$
$$g_{t+1}^T \theta - g_{t+1}^T \mu_t - \frac{1}{2}\theta^T \Sigma_t^{-1}(\gamma_t)\theta + \theta^T \Sigma_t^{-1}(\gamma_t)\mu_t(\gamma_t) - \frac{1}{2}\mu_t(\gamma_t)^T \Sigma_t^{-1}(\gamma_t)\mu_t(\gamma_t) =$$
$$-\frac{1}{2}\theta^T \Sigma_t^{-1}(\gamma_t)\theta + \theta^T(\Sigma_t^{-1}(\gamma_t)\mu_t(\gamma_t) + g_{t+1}) - g_{t+1}^T \mu_t - \frac{1}{2}\mu_t(\gamma_t)^T \Sigma_t^{-1}(\gamma_t)\mu_t(\gamma_t) =$$
$$-\frac{1}{2}\left(\theta^T \Sigma_t^{-1}\theta - 2\theta^T(\Sigma_t^{-1}(\gamma_t)\mu_t(\gamma_t) + g_{t+1})\right) - g_{t+1}^T \mu_t - \frac{1}{2}\mu_t(\gamma_t)^T \Sigma_t^{-1}(\gamma_t)\mu_t(\gamma_t)$$

Let's focus on this term
$$-\frac{1}{2}\left(\theta^T \Sigma_t^{-1}\theta - 2\theta^T(\Sigma_t^{-1}(\gamma_t)\mu_t(\gamma_t) + g_{t+1})\right) =$$
$$-\frac{1}{2}\left(\theta^T \Sigma_t^{-1}\theta - 2\theta^T \Sigma_t^{-1}b(\gamma_t)\right) =$$
$$-\frac{1}{2}\left(\theta^T \Sigma_t^{-1}\theta - 2\theta^T \Sigma_t^{-1}b(\gamma_t) + b(\gamma_t)^T \Sigma_t^{-1}b(\gamma_t) - b(\gamma_t)^T \Sigma_t^{-1}b(\gamma_t)\right) =$$
$$-\frac{1}{2}\left(\theta^T \Sigma_t^{-1}\theta - 2\theta^T \Sigma_t^{-1}b(\gamma_t) + b(\gamma_t)^T \Sigma_t^{-1}b(\gamma_t)\right) + \frac{1}{2}b(\gamma_t)^T \Sigma_t^{-1}b(\gamma_t) =$$
$$-\frac{1}{2}(\theta - b(\gamma_t))^T \Sigma_t^{-1}(\theta - b(\gamma_t)) + \frac{1}{2}b(\gamma_t)^T \Sigma_t^{-1}b(\gamma_t)$$

where

$$b(\gamma_t) = \Sigma_t(\gamma_t)\left[\Sigma_t^{-1}(\gamma_t)\mu_t(\gamma_t) + g_{t+1}\right]$$

Therefore, the integral could be written as

$$\log \int \exp^{g_{t+1}^T(\theta - \mu_t)} \exp^{-\frac{1}{2}(\theta - \mu_t(\gamma_t))^T \Sigma_t^{-1}(\gamma_t)(\theta - \mu_t(\gamma_t))} d\theta \frac{1}{\sqrt{(2\pi)^D |\Sigma_t(\gamma_t)|}} =$$

$$\frac{1}{2}b(\gamma_t)^T \Sigma_t^{-1} b(\gamma_t) + \log \int \exp^{-\frac{1}{2}(\theta - b(\gamma_t))^T \Sigma_t^{-1}(\theta - b(\gamma_t))} d\theta \frac{1}{\sqrt{(2\pi)^D |\Sigma_t(\gamma_t)|}} - g_{t+1}^T \mu_t - \frac{1}{2}\mu_t(\gamma_t)^T \Sigma_t^{-1}(\gamma_t)\mu_t(\gamma_t) =$$

$$\frac{1}{2}b(\gamma_t)^T \Sigma_t^{-1}(\gamma_t) b(\gamma_t) - g_{t+1}^T \mu_t - \frac{1}{2}\mu_t(\gamma_t)^T \Sigma_t^{-1}(\gamma_t)\mu_t(\gamma_t)$$

Now, we only keep the terms depending on $\gamma_t$

$$\frac{1}{2}b(\gamma_t)^T \Sigma_t^{-1}(\gamma_t) b(\gamma_t) - \frac{1}{2}\mu_t(\gamma_t)^T \Sigma_t^{-1}(\gamma_t)\mu_t(\gamma_t) =$$

$$\frac{1}{2}\left[\Sigma_t^{-1}(\gamma_t)\mu_t(\gamma_t) + g_{t+1}\right]^T \Sigma_t(\gamma_t)\left[\Sigma_t^{-1}(\gamma_t)\mu_t(\gamma_t) + g_{t+1}\right] - \frac{1}{2}\mu_t(\gamma_t)^T \Sigma_t^{-1}(\gamma_t)\mu_t(\gamma_t) =$$

$$\frac{1}{2}\mu_t(\gamma_t)^T \Sigma_t^{-1}(\gamma_t)\mu_t(\gamma_t) + g_{t+1}^T \mu_t(\gamma_t) + g_{t+1}^T \frac{1}{2}\Sigma_t(\gamma_t)g_{t+1} - \frac{1}{2}\mu_t(\gamma_t)^T \Sigma_t^{-1}(\gamma_t)\mu_t(\gamma_t) =$$

$$g_{t+1}^T \mu_t(\gamma_t) + \frac{1}{2}g_{t+1}^T \Sigma_t(\gamma_t)g_{t+1}$$

Since $\Sigma_t(\gamma_t) = diag(\sigma_t^2 \circ \gamma_t^2 + (1 - \gamma_t^2) \circ \sigma_0^2)$, we recover

$$g_{t+1}^T(\gamma_t \circ \mu_t + (1 - \gamma_t) \circ \mu_0) + \frac{1}{2}g_{t+1}^T\left((\sigma_t^2 \circ \gamma_t^2 + (1 - \gamma_t^2) \circ \sigma_0^2) \circ g_{t+1}\right)$$

Now, we add an l2-penalty $\frac{\lambda}{2}||\gamma_t - \gamma_{t,0}||^2$ and we get

$$F(\gamma_t) = g_{t+1}^T(\gamma_t \circ \mu_t + (1 - \gamma_t) \circ \mu_0) + \frac{1}{2}g_{t+1}^T\left((\sigma_t^2 \circ \gamma_t^2 + (1 - \gamma_t^2) \circ \sigma_0^2) \circ g_{t+1}\right) - \frac{\lambda}{2}||\gamma_t - \gamma_{t,0}||^2$$

Let's take the gradient wrt $\gamma_t^j$, we get that the $j$-the component of the gradient is given by

$$(\nabla F(\gamma_t))^j = g_{t+1}^j(\mu_t^j - \mu_0^j) + g_{t+1}^j\left((\sigma_t^{j2}\gamma_t^j - \sigma_0^{j2}\gamma_t^j)g_{t+1}^j\right) - \lambda(\gamma_t^j - \gamma_{t,0}^j) =$$

$$g_{t+1}^j(\mu_t^j - \mu_0^j) + \lambda\gamma_{t,0}^j - \gamma_t^j\left[\lambda + g_{t+1}^j\left((\sigma_0^{j2} - \sigma_t^{j2}) \circ g_{t+1}^j\right)\right] = 0 \qquad (49)$$

We have then,

$$\gamma_t^j = \frac{g_{t+1}^j(\mu_t^j - \mu_0^j) + \lambda\gamma_{t,0}^j}{\lambda + g_{t+1}^{j}{}^2(\sigma_0^{j2} - \sigma_t^{j2})}$$

In case when $\gamma_t$ are shared for a subset of $K$ parameters, indexes by the index set $J_K = (j_1, \ldots, j_K)$, then, we can sum the gradients (49) and we get

$$\sum_{j \in J_K} g_{t+1}^j(\mu_t^j - \mu_0^j) + \lambda\gamma_{t,0} - \gamma_t\left[\lambda + g_{t+1}^j\left((\sigma_0^{j2} - \sigma_t^{j2}) \circ g_{t+1}^j\right)\right] = 0,$$

this gives us

$$\gamma_t = \frac{\bar{g}_{t+1}^T(\bar{\mu}_t - \bar{\mu}_0) + K\lambda\gamma_{t,0}}{(\bar{g}_{t+1} \circ (\bar{\sigma}_0^2 - \bar{\sigma}_t^2))^T\bar{g}_{t+1} + K\lambda},$$

where $\bar{a}$ is vector defined on the index set $J_K$, i.e., $\bar{a} = (a_{j_1}, \ldots, a_{j_K})$.

# G    Proximal SGD

Each step of online SGD can be seen in terms of a regularized minimization problem referred to as the proximal form [44]:

$$\hat{\theta}_{t+1} = \arg\min_\theta \widehat{\mathcal{L}}_{t+1}(\theta) + \frac{1}{2\alpha_t}||\theta - \theta_t||^2. \tag{50}$$

In general, we cannot solve (50) directly, so we consider a Taylor expansion of $\widehat{\mathcal{L}}_{t+1}$ around $\theta_t$, giving

$$\theta_{t+1} = \arg\min_\theta \nabla_\theta \widehat{\mathcal{L}}_{t+1}(\theta_t)^\top(\theta - \theta_t) + \frac{1}{2\alpha_t}||\theta_t - \theta||^2. \tag{51}$$

Here we see the role of $\alpha_t > 0$ as both enforcing that the Taylor expansion around $\theta_t$ is accurate, and regularising $\theta_{t+1}$ towards the old parameters $\theta_t$ (hence ensuring that the learning from past data is not forgotten). Solving (51) naturally leads to the well known SGD update:

$$\theta_{t+1} = \theta_t - \alpha_t \nabla_\theta \widehat{\mathcal{L}}_{t+1}(\theta_t),$$

where $\alpha_t$ can now also be interpreted as the learning rate.

# H    Experimental details

## H.1    Plasticity experiments

**Tasks**    In this section we provide experimental details. As plasticity tasks, we use a randomly selected subset of size $10000$ from CIFAR-10 [32] and from MNIST. This subset is fixed for all the tasks. Within each task, we randomly permute labels for every image; we call such problems random-label classification problems. We study two regimes – *data efficient*, where we do $400$ epochs on a task with a batch size of $128$, and *memorization*, a regime where we do only $70$ epochs with a batch size of $128$. As the main backbone architecture, we use MLP with $4$ hidden layers each having a hidden dimension of $256$ hidden units. We use ReLU activation function and do not use any batch or layer normalization. For the incoming data, we apply random crop, for MNIST to produce images of size $24 \times 24$ and for CIFAR-10 to produce images of size $28 \times 28$. We normalize images to be within $[0, 1]$ range by dividing by $255$. On top of that, we consider *permuted MNIST* task with a similar training ragime as in [34] – we consider a subset of $10000$ images, with batch size $16$ and each task is one epoch. As a backbone, we still use MLP with ReLU activation and $4$ hidden layers. Moreover, we considered *permuted Patch MNIST*, where we permute patches, not individual pixels. In this case, we used a simple $4$ layer convolutional neural network with $2$ fully connected layers at the end.

**Metrics**    We use *online accuracy* as first metric with results reported in Appendix K. Moreover we use *per-task Average Online Accuracy* which is

$$\mathcal{A}_t = \frac{1}{N} \sum_{i=1}^{N} a_i^t, \tag{52}$$

where $a_i^t$ are the online accuracies collected on the task $t$ via $N$ timesteps.

**Baselines**    First baseline is *Online SGD* which sequentially learns over the sequence of task, with a fixed learning rate. *Hard Reset* is the *Online SGD* which resets all the parameters at task boundaries. *L2 init* [34] adds a regularizer $\lambda||\theta - \theta_0||^2$ term to each *Online SGD* update where the regularization strength $\lambda$ is a hyperparameter. *Shrink & Perturb* applies the transformation $\lambda\theta_t + \sigma\epsilon, \epsilon \sim \mathcal{N}(\epsilon; 0, I)$ to each parameter before the gradient update. The hyperparameters are $\lambda$ and $\sigma$.

**Soft Reset**    corresponds to one update (10) starting from 1 using 1 Monte Carlo estimate. We always use 1 Monte Carlo estimate for updating $\gamma_t$ as we found that it worked well in practice on these tasks. The hyperparameters of the method – $\sigma_0^2$ initial variance of the prior, which we set to be equal to $\nu^2 \frac{1}{N}$ where $N$ is the width of the hidden layer and $\nu$ is a constant (hyperparameter). It always equals to $\nu = 0.1$. On top of that the second hyperparameter is $s$, such that $\sigma_t = s\sigma_0$, which controls the relative decrease of the constant posterior variance. This is the hyperparameter over which we sweep

over. Another hyperparameter is the learning rate for learning $\gamma_t$. For *Soft Reset Proximal*, we also have a proximal coefficient regularization constant $\lambda$. Besides that, we also sweep over the learning rate for the parameter. For the *Bayesian Soft Reset*, we just add an additional learning rate for the variance $\alpha_\sigma$ and we do 1 Monte Carlo sample for each ELBO update.

**Hyper parameters selection and evaluation**   For all the experiments, we run a sweep over the hyperparameters. We select the best hyperparameters based on the smallest cumulative error (sum of all $1 - a_i^t$ throughout the training). We then report the mean and the standard deviation across 3 seeds in all the plots.

**Hyperparameter ranges**   . Learning rate $\alpha$ which is used to update parameters, for all the methods, is selected from $\{1e-4, 5e-4, 1e-3, 5e-3, 1e-2, 5e-2, 1e-1, 5e-1, 1.0\}$. The $\lambda_{init}$ parameter in *L2 Init*, is selected from $\{10.0, 1.0, 0.0, 1e-1, 5e-1, 1e-2, 5e-2, 1e-3, 5e-3, 1e-4, 5e-4, 1e-5, 5e-5, 1e-6, 5e-6, 1e-7, 5e-7, 1e-8, 5e-8, 1e-9, 5e-9, 1e-10, \}$. For S&P, the shrink parameter $\lambda$ is selected from $\{1.0, 0.99999, 0.9999, 0.999, 0.99, 0.9, 0.8, 0.7, 0.5, 0.3, 0.2, 0.1\}$, and the perturbation parameter $\sigma$ is from $\{1e-1, 1e-2, 1e-3, 1e-4, 1e-5, 1e-6\}$. As noise distribution, we use the Neural Network initial distribution. For *Soft Resets*, the learning rate for $\gamma_t$ is selected from $\{0.5, 0.1, 0.05, 0.01, 0.005, 0.001, 0.0005, 0.0001\}$, the constant $s$ is selected from $\{1.0, 0.95, 0.9, 0.8, 0.7, 0.6, 0.5, 0.3, 0.1\}$, the temperature $\lambda$ in (45) is selected from $\{1.0, 0.1, 0.01\}$, the same is true for the temperature in the Bayesian method (38). Initial prior std rescaling $\nu = 0.05$. On top of that for the Bayesian method, we always use $\nu$ (see Algorithm 2) equal to $\nu = 0.05$ and $\pi = 0.9$, i.e. the posterior is always slightly smaller than the prior. Finally for the Bayesian method we had to learn the variance with learning rate from $\{0.01, 0.1, 1, 10\}$ range.

In practice, we found that there is one learning rate of $0.1$, which was always the best in practice for most of the methods and only proximal *Soft Resets* on *memorization* CIFAR-10 required smaller learning rate $0.01$. This allowed us to significantly reduce the hyperparameter sweep.

## H.2   Impact of non-stationarity experiments

In this experiment, we consider a subset of $10000$ images from MNIST (fixed throughtout all experiment) and a sequence of tasks. Each task is constructed by assigning either a true or a random label to each image from MNIST, where the probability of assignment is controlled by the experiment. The duration of each is controlled by the number of epochs with batch size of $128$. As backbone we use MLP with $4$ hidden layers and $256$ hidden units and ReLU activation. For all the methods, the learning rate is $0.1$. For *Soft Resets*, we use $s = 0.9$ and $\nu = 1$ and $\eta_\gamma = 0.01$. Bayesian method uses temperature $\lambda = 0.01$. Detailed results are given in Figure 7.

## H.3   Reinforcement learning experiments

We conduct experiments in the RL environments. We take the canonical implementation of Soft-Actor Critic(SAC) from Brax [15] repo in github, which uses 2 layer MLPs for both policy and Q-function. It employs ReLU activation functions for both. On top of that, it uses 2 MLP networks to parameterize $Q$-function (see Brax [15]) for more details. To employ *Soft Reset*, we do the following. After we have collected a chunk of data (128) time-steps, we do one update (10) on $\gamma_t$ starting from 1 at every update of $\gamma_t$, where $\gamma_t$ is shared for all the parameters within each layer of a Neural Network, separately for weights and biases. On top of that, since we have policy and value function networks, we have separate $\gamma_t$ for each of these. After the update on $\gamma_t$, we compute $\theta_t(\gamma_t)$ and $\alpha_t(\gamma_t)$, see Section 3.5. After that, we employ the proximal objective (45) with a fixed regularization target $\theta_t(\gamma_t)$. Concretely, we use the update rule (47) where for each update the gradient is estimate on the batch of data from the replay buffer. This is not exactly the same as what we did with *plasticity benchmarks* since there the update was applied to the same batch of data, multiple times. Nevertheless, we found this strategy effective and easy to implement on top of a SAC algorithm. In practice, we swept over the parameter $s$ (similar for both, policy and the value function) which controls the relative learning rate increase in (19). Moreover, we swept over the temperature $\tilde{\lambda}$ from eqn. (45), which was different for the policy and for the value function. In practice, we found that using temperature of $0$ for the policy led to the best empirical results. The range for the temperatures $\tilde{\lambda}$ was $\{0.1, 0.01, 0.001\}$ and for $s$ was $\{0.8, 0.9, 0.95, 0.97, 1.0\}$. We used $\nu = 1$ for all the experiments. For each experiment, we used a 3 hours of the $A100$ GPU with 40 Gb of memory.

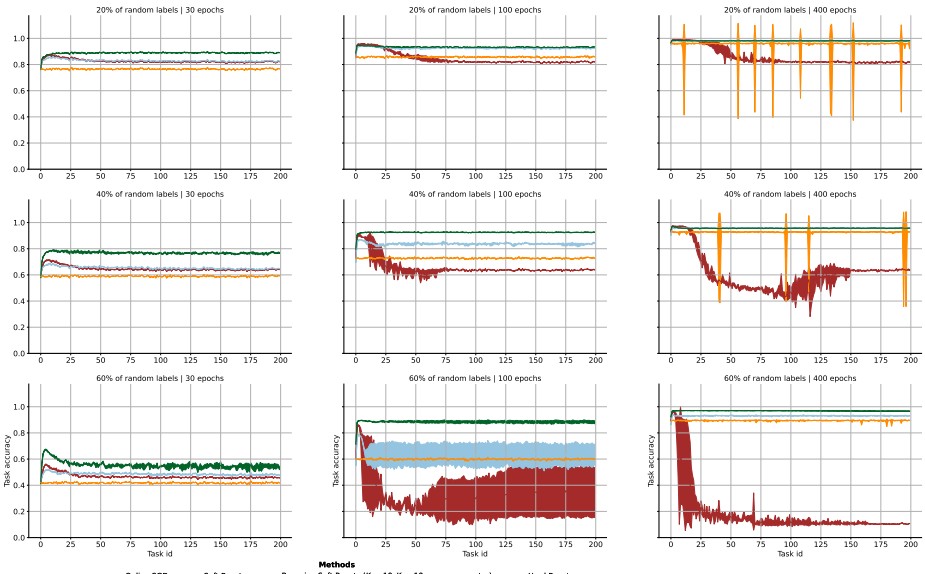

Figure 7: **Non-stationarity impact**. The x-axis denotes task id, each column denotes the duration, whereas a row denotes the amount of label noise. Each color denotes the method studied. The y-axis denotes average over 3 seeds online accuracy.

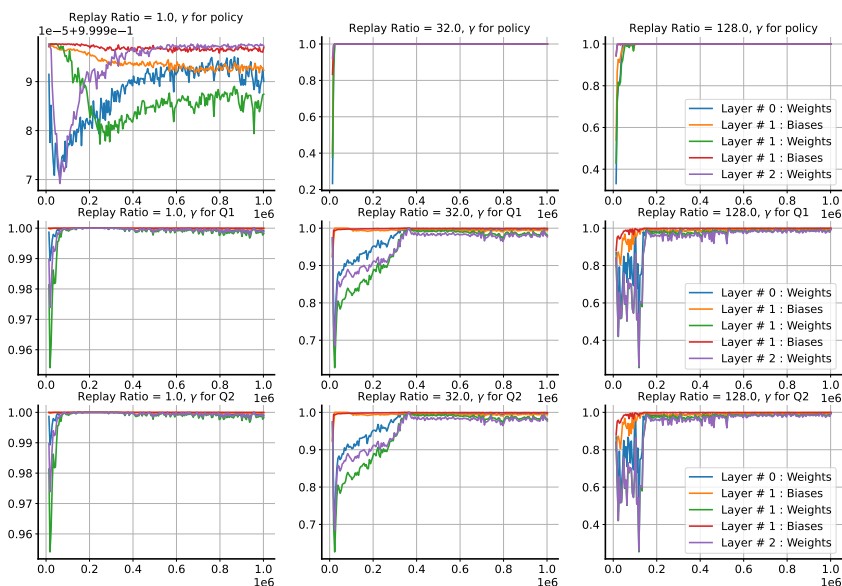

Figure 8: Visualization of the $\gamma_t$ dynamics for the run on Humanoid environment. Each column corresponds to the replay ratio studied. First row denotes the $\gamma_t$ for the policy $\pi$. The second and the third rows denote the $\gamma_t$ for the two $Q$-functions.

# I   Computational complexity

We provide the study of computational cost for all the proposed methods. Notations:

- $P$ be the number of parameters in the Neural Network
- $L$ is the number of layers
- $O(S)$ is the cost of SGD backward pass.
- $M_\gamma$ - number of Monte Carlo samples for the drift model

- $M_\theta$ - number of Monte Carlo samples for the parameter updates (Bayesian Method).

- $K_\gamma$ - number of updates for the drift parameter

- $K_\theta$ - number of NN parameter updates.

| Method | Comp. cost | Memory |
|---|---|---|
| SGD | $O(S)$ | $O(P)$ |
| Soft resets $\gamma$ per layer | $O(K_\gamma M_\gamma S + S)$ | $O(L + (M_\gamma + 1)P)$ |
| Soft resets $\gamma$ per param. | $O(K_\gamma M_\gamma S + S)$ | $O(P + (M_\gamma + 1)P)$ |
| Soft resets $\gamma$ per layer + proximal ($K_\theta$ iters) | $O(K_\gamma M_\gamma S + K_\theta S)$ | $O(L + (M_\gamma + 1)P)$ |
| Soft resets $\gamma$ per param. + proximal ($K_\theta$ iters) | $O(K_\gamma M_\gamma S + K_\theta S)$ | $O(P + (M_\gamma + 1)P)$ |
| Bayesian Soft Reset Proximal ($K_\theta$ iters) $\gamma$ per layer | $O(K_\gamma M_\gamma S + 2M_\theta K_\theta S)$ | $P(L + (M_\gamma + 2)P)$ |
| Bayesian Soft Reset Proximal ($K_\theta$ iters) $\gamma$ per param. | $O(K_\gamma M_\gamma S + 2M_\theta K_\theta S)$ | $P(P + (M_\gamma + 2)P)$ |

Table 1: Comparison of methods, computational cost, and memory requirements

The general theoretical cost of all the proposed approaches is given in Table 1. In practice, for all the experiments, we assume that $M_\gamma = 1$ and $M_\theta = 1$. Moreover, we used $K_\gamma = 1$ and $K_\theta = 1$ for *Soft Reset*, $K_\gamma = 10$ and $K_\theta = 1$ for *Soft Reset* with more computation. On top of that, for *Soft Reset* proximal and all Bayesian methods, we used $K_\gamma = 10$ and $K_\theta = 10$. Table 2, quantifying the complexity of all the methods from Figure 2.

| Method | Comp. cost | Memory |
|---|---|---|
| SGD | $O(S)$ | $O(P)$ |
| Soft resets $\gamma$ per layer | $O(2S)$ | $O(L + 2P)$ |
| Soft resets $\gamma$ per param. | $O(2S)$ | $O(3P)$ |
| Soft resets $\gamma$ per layer + proximal ($K_\theta = 10$ iters) | $O(20S)$ | $O(L + 2P)$ |
| Soft resets $\gamma$ per param. + proximal ($K_\theta$ iters) | $O(20S)$ | $O(3P)$ |
| Bayesian Soft Reset Proximal ($K_\theta$ iters) $\gamma$ per layer | $O(30S)$ | $P(L + 3P)$ |
| Bayesian Soft Reset Proximal ($K_\theta$ iters) $\gamma$ per param. | $O(30S)$ | $P(4)$ |

Table 2: Comparison of methods, computational cost, and memory requirements for methods in Figure 2.

The complexity $O(2S)$ of Soft Resets comes from one update on drift parameter and one updat eon NN parameters. The memory complexity requires storing $O(L)$ parameters gamma (one for each layer), parameters $\theta_t$ with $O(P)$ and sampled parameters for drift model update which requires $O(P)$.

Note that as Figure 9 suggests, it is beneficial to spend more computational cost on optimizing gamma and on doing multiple updates on parameters. However, even the cheapest version of our method *Soft Resets* still leads to a good performance as indicated in Figure 2.

The complexity of soft resets in reinforcement learning setting requires only one gradient update on $\gamma$ after each new chunk of fresh data from the environment. In SAC, we do $G$ gradient updates on parameters for every new chunk of data. Assuming that complexity of one gradient update in SAC is $O(S)$, soft reset only requires doing one additional gradient update to fit $\gamma$ parameter.

| Method | Comp. cost | Memory |
|---|---|---|
| SAC | $O(GS)$ | $O(P)$ |
| Soft resets $\gamma$ per layer | $O(S + GS)$ | $O(L + 2P)$ |

Table 3: Comparison of methods, computational cost, and memory requirements for methods in for RL.

The computation complexity of Soft Reset in Reinforcement Learning is marginally higher than SAC but leads to better empirical performance in a highly off-policy regime, see Appendix H.3.

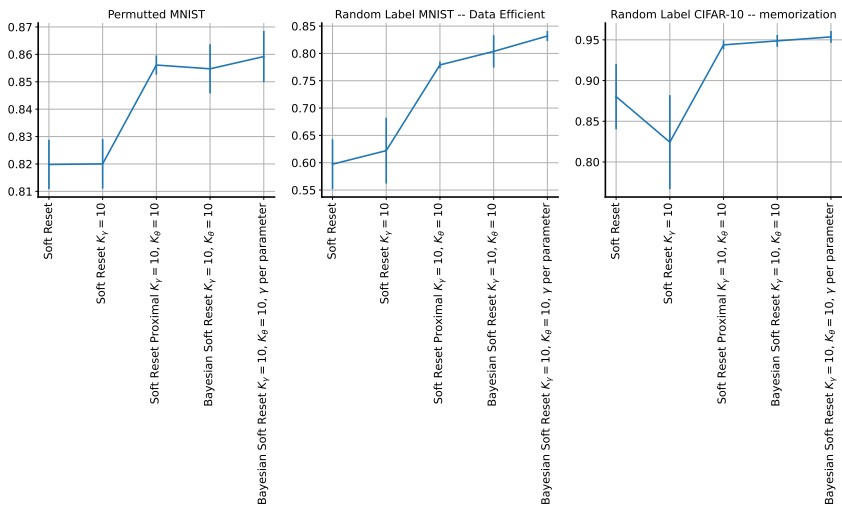

Figure 9: Compute-performance tradeoff. The x-axis indicates the method going from the cheapest (left) to the most expensive (right). See Table 2 for complexity analysis. The y-axis is the average performance on all the tasks across the stream.

## J   Sensitivity analysis

We study the sensitivity of Soft Resets where $\gamma$ is defined per layer when trained on random-label MNIST (data efficient). We fix the learning rate to $\alpha = 0.1$. We study the sensitivity of learning rate for the drift parameter, $\eta_\gamma$, as well as $\nu$ – initial prior standard deviation rescaling, and $s$ – posterior standard deviation rescaling parameter.

On top of that, we conduct the sensitivity analysis of L2 Init [34] and Shrink&Perturb [2] methods. The x-axis of each plot denotes one of the studied hyperparameters, whereas y-axis is the average performance across all the tasks (see Experiments section for tasks definition). The standard deviation is reported over 3 random seeds. A color indicates a second hyperparameter which is studied, if available. In the title of each plot, we write hyperparameters which are fixed. The analysis is provided in Figure 10 for *Soft Resets* and in Figure 11 for the baselines.

The most important parameter is the learning rate of the drift model $\eta_\gamma$. For each method, there exists a good value of this parameter and performance is sensitive to it. This makes sense since this parameter directly impacts how we learn the drift model.

The performance of Soft Resets is robust with respect to the posterior standard deviation scaling $s$ parameter as long as it is $s \geq 0.5$. For $s < 0.5$, the performance degrades. This parameter is defined from $\sigma_t = s\sigma_0$ and affects relative increase in learning rate given by $\frac{1}{\gamma^2 + (1-\gamma^2)/s^2}$ which could be ill-behaved for small $s$.

We also study the sensitivity of the baseline methods. We find that L2 Init [34] is very sensitive to the parameter $\lambda$, which is a penalty term for $\lambda||\theta - \theta_0||^2$. In fact, Figure 11, left shows that there is only one good value of this parameter which works. Shrink&Perturb [2] is very sensitive to the shrink parameter $\lambda$. Similar to L2 Init, there is only one value which works, 0.9999 while values 0.999 and values 0.99999 lead to bad performance. This method however, is not very sensitive to the perturb parameter $\sigma$ provided that $\sigma \leq 0.001$.

Compared to the baselines, our method is more robust to the hyperparameters choice. Below, we also add sensitivity analysis for other method variants. Figure 12 shows sensitivity of *Soft Resets*, $K_\gamma = 10$, Figure 13 shows sensitivity of *Soft Resets*, $K_\gamma = 10$, $K_\theta = 10$, Figure 14 shows sensitivity of *Bayesian Soft Resets*, $K_\gamma = 10$, $K_\theta = 10$ with $\gamma_t$ per layer, Figure 15 shows sensitivity of *Bayesian Soft Resets*, $K_\gamma = 10$, $K_\theta = 10$ with $\gamma_t$ per parameter.

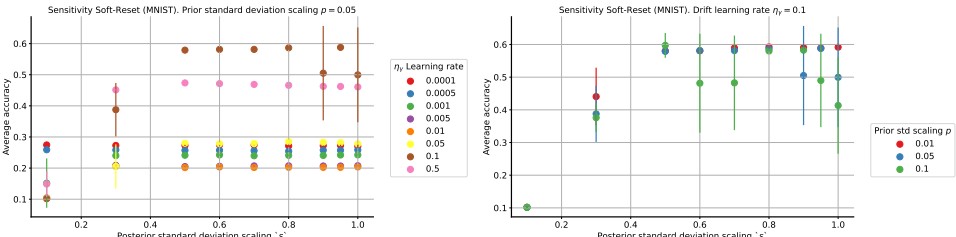

Figure 10: **Soft Reset**, **sensitivity analysis** of performance with respect to the hyperparameters on data-efficient random-label MNIST. The x-axis denotes the studied hyperparameter, whereas the y-axis denotes the average performance across the tasks. The standard deviation is computed over 3 random seeds. The color indicates additional studied hyperparameter. **(Left)** shows sensitivity analysis where the x-axis is the posterior standard deviation scaling $s$ and the color indicates the drift model learning rate $\eta_\gamma$. **(Right)** shows sensitivity of *Soft Reset* where the x-axis is the posterior standard deviation scaling $s$ and the color indicates initial prior standard deviation scaling $\nu$.

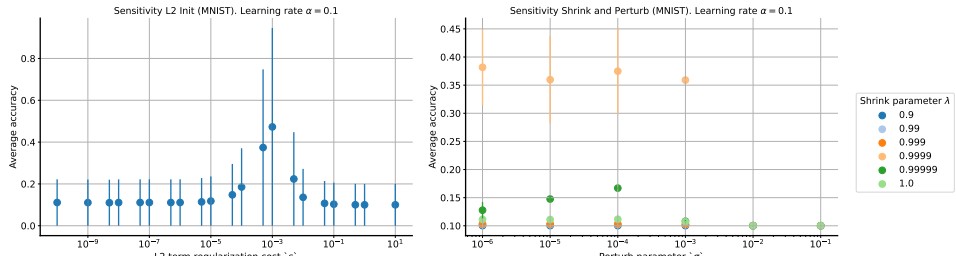

Figure 11: **L2 Init and Shrink&Perturb sensitivity analysis** of performance with respect to the hyperparameters on data-efficient random-label MNIST. The x-axis denotes the studied hyperparameter, whereas the y-axis denotes the average performance across the tasks. The standard deviation is computed over 3 random seeds. The color optionally indicates additional studied hyperparameter. **(Left)** shows sensitivity of *L2 Init* with respect to the $L2$ penalty regularization cost $\lambda$ applied to $||\theta - \theta_0||^2$ term. We do not use an additional hyperparameter, therefore there is only one color. **(Right)** shows sensitivity of *Shrink&Perturb* method where the x-axis is the perturb parameter $\sigma$ while the color indicates the shrink parameter $\lambda$.

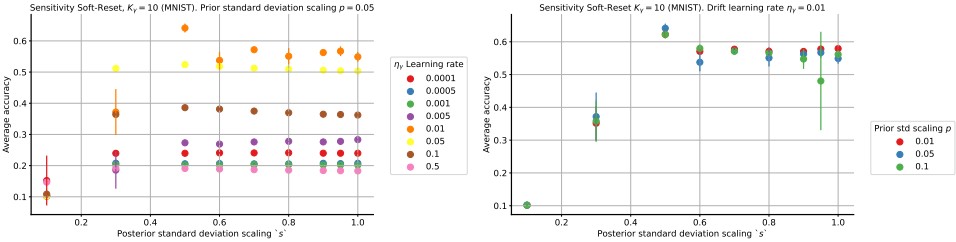

Figure 12: **Soft Reset**, $K_\gamma = 10$, **sensitivity analysis** of performance with respect to the hyperparameters on data-efficient random-label MNIST. The x-axis denotes the studied hyperparameter, whereas the y-axis denotes the average performance across the tasks. The standard deviation is computed over 3 random seeds. The color indicates additional studied hyperparameter. **(Left)** shows sensitivity analysis where the x-axis is the posterior standard deviation scaling $s$ and the color indicates the drift model learning rate $\eta_\gamma$. **(Right)** shows sensitivity analysis where the x-axis is the posterior standard deviation scaling $s$ and the color indicates initial prior standard deviation scaling $\nu$.

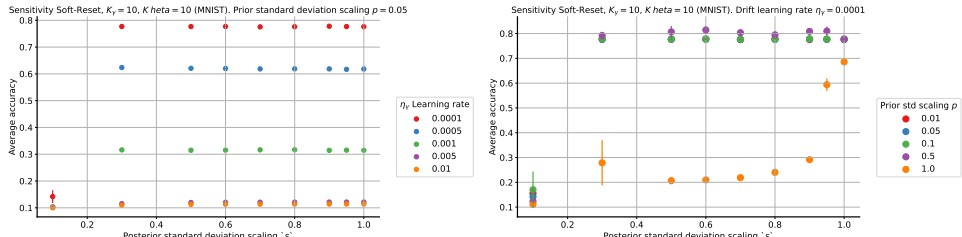

Figure 13: **Soft Reset**, $K_\gamma = 10, K_\theta = 10$, **sensitivity analysis** of performance with respect to the hyperparameters on data-efficient random-label MNIST. The x-axis denotes the studied hyperparameter, whereas the y-axis denotes the average performance across the tasks. The standard deviation is computed over 3 random seeds. The color indicates additional studied hyperparameter. **(Left)** shows sensitivity analysis where the x-axis is the posterior standard deviation scaling $s$ and the color indicates the drift model learning rate $\eta_\gamma$. **(Right)** shows sensitivity analysis where the x-axis is the posterior standard deviation scaling $s$ and the color indicates initial prior standard deviation scaling $\nu$.

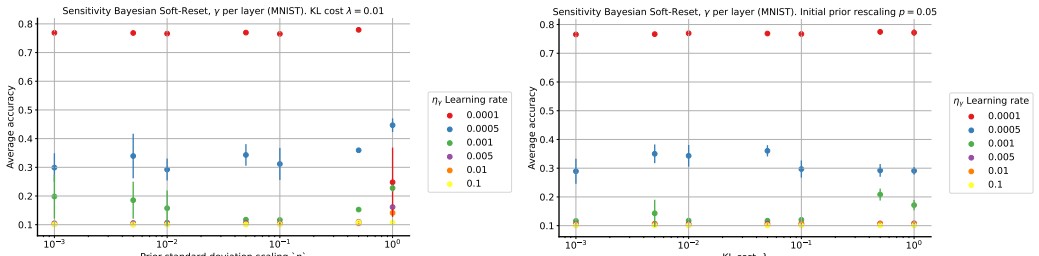

Figure 14: **Bayesian Soft Reset**, $K_\gamma = 10, K_\theta = 10$ **with $\gamma_t$ per layer, sensitivity analysis** of performance with respect to the hyperparameters on data-efficient random-label MNIST. The x-axis denotes the studied hyperparameter, whereas the y-axis denotes the average performance across the tasks. The standard deviation is computed over 3 random seeds. The color indicates additional studied hyperparameter. **(Left)** shows sensitivity analysis where the x-axis is the prior standard deviation initial scaling $\nu$ and the color indicates the drift model learning rate $\eta_\gamma$. **(Right)** shows sensitivity analysis where the x-axis is the KL divergence coefficient $\lambda$ while the color indicates the learning rate $\eta_\gamma$.

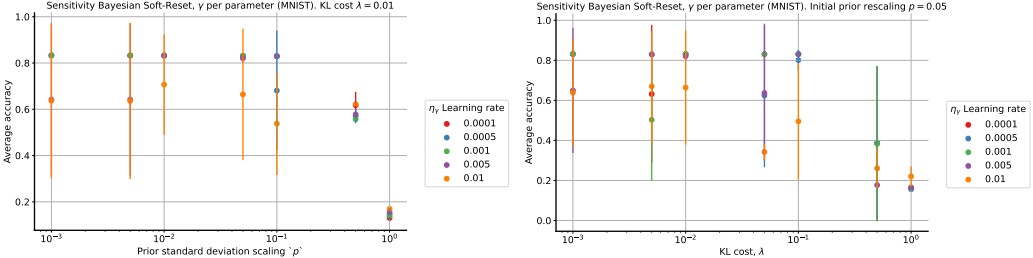

Figure 15: **Bayesian Soft Reset**, $K_\gamma = 10, K_\theta = 10$ **with $\gamma_t$ per parameter, sensitivity analysis** of performance with respect to the hyperparameters on data-efficient random-label MNIST. The x-axis denotes the studied hyperparameter, whereas the y-axis denotes the average performance across the tasks. The standard deviation is computed over 3 random seeds. The color indicates additional studied hyperparameter. **(Left)** shows sensitivity analysis where the x-axis is the prior standard deviation initial scaling $\nu$ and the color indicates the drift model learning rate $\eta_\gamma$. **(Right)** shows sensitivity analysis where the x-axis is the KL divergence coefficient $\lambda$ while the color indicates the learning rate $\eta_\gamma$.

# K    Qualitative behavior of soft resets and additional results on Plasticity benchmarks

## K.1    Perfect Soft Resets

To understand the impact of drift model (5), we study the *data efficient* random-label MNIST setting where task boundaries are known. We run *Online SGD*, *Hard Reset* which resets all parameters at task

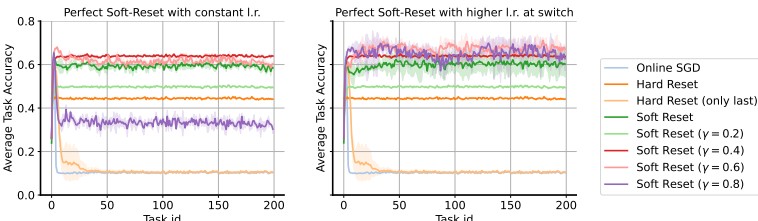

Figure 16: **Perfect soft-resets** on *data-efficient* random-label MNIST. *Left*, *Soft Reset* method does not use higher learning rate when $\gamma < 1$. *Right*, *Soft Reset* increases the learning rate when $\gamma < 1$, see (19). The x-axis represents task id, whereas the y-axis is the average training accuracy on the task.

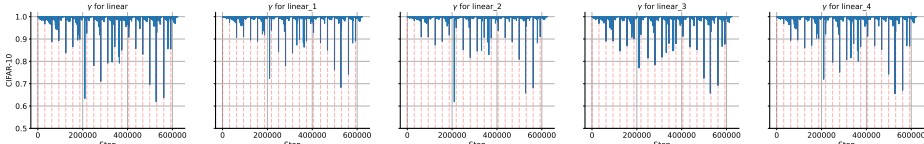

Figure 17: Behaviour of $\gamma_t$ for different layers on random-label MNIST (data efficient) for the first 20 tasks.

boundaries, and *Hard Reset (only last)* which resets only the last layer. We use *Soft Reset* method (20) where $\gamma_t = 1$ all the time and becomes $\gamma_t = \hat{\gamma}_t$ (with manually chosen $\hat{\gamma}_t$) at task boundaries. We consider constant learning rate $\alpha_t(\gamma_t)$ and increasing learning rate (19) at task boundary for *Soft Reset*. On top of that, we run *Soft Reset* method unaware of task boundaries which learns $\gamma_t$. We report *Average training task accuracy* metric in Figure 16. See Appendix H.1 for details. The results suggest that with the appropriate choice of $\hat{\gamma}_t$, *Soft Reset* is much more efficient than *Hard Reset* and the effect becomes stronger if the learning rate $\alpha_t(\gamma_t)$ increases. We also see that *Soft Reset* could learn an appropriate $\gamma_t$ without the knowledge of task boundary.

## K.2   Qualitative Behaviour on *Soft Resets* on random-label tasks.

We observe what values of $\gamma_t$ we get as we train *Soft Reset* method on random-label MNIST (data-efficient) and CIFAR-10 (memorization). The results are given in Figure 17 for MNIST and in Figure 18 for CIFAR-10. We report these for the first 20 tasks.

## K.3   Qualitative Behaviour on *Soft Resets* on permuted patches of MNIST.

We consider a version of permuted MNIST where instead of permuting all the pixels, we permute patches of pixels with a patch size varying from 1 to 14. The patch size of 1 corresponds to permututed MNIST and therefore the most non-stationary case, while patch size of 14 corresponds to least non-stationary case. We use a convolutional Neural Network in this case. In Figure 19, we report the behavior of $\gamma$ for different convolutional and fully connected layers on first few tasks.

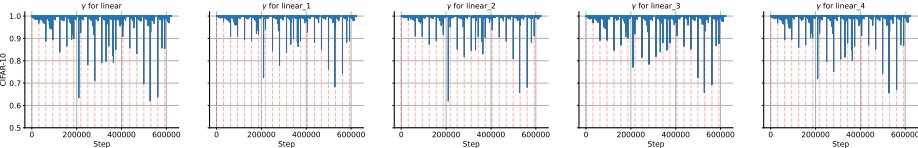

Figure 18: Behaviour of $\gamma_t$ for different layers on random-label CIFAR-10 (memorization) for the first 20 tasks.

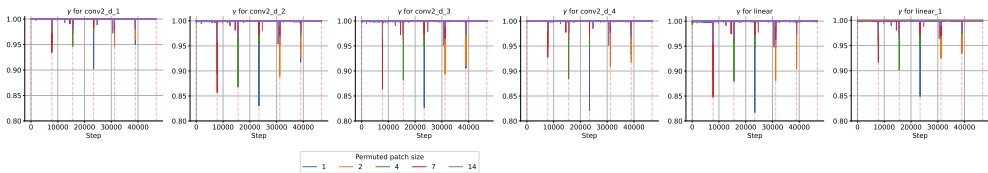

Figure 19: Behaviour of $\gamma_t$ for different layers on permuted MNIST

### K.4 Bayesian method is better than non-Bayesian

As discussed in Section 5, we found that in practice *Soft Reset* and *Soft Reset Proximal* where $\gamma$ is learned per-parameter, did not perform well on the plasticity benchmarks. However, the Bayesian variant described in Section E.1, actually benefited from specifying $\gamma$ for every parameter in Neural Network. We report these additional results in Figure 20. We see that the non Bayesian variants where $\gamma_t$ is specified per parameter, do not perform well. The fact that the Bayesian method performs better here suggests that it is important to have a good uncertainty estimate $\sigma_t^2$ for the update (10) on $\gamma_t$. When, however, we regularize $\gamma_t$ to be shared across all parameters within each layer, this introduces useful inductive bias which mitigates the lack of uncertainty estimation in the parameters. This is because for non-Bayesian methods, we assume that the uncertainty is fixed, given by a hyperparameter – assumption which would not always hold in practice.

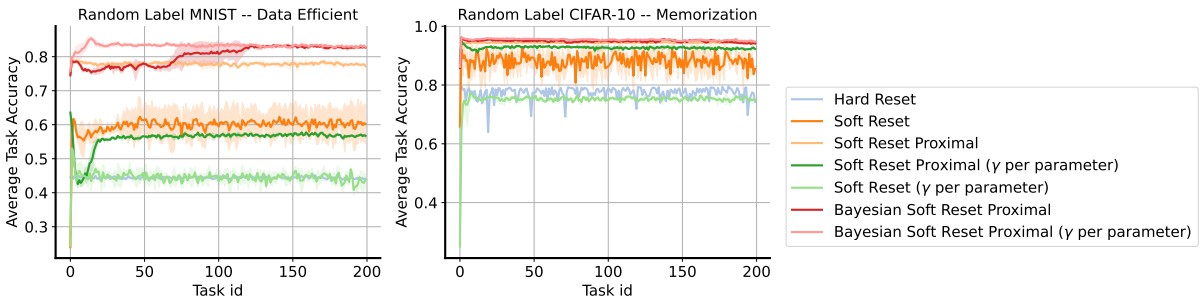

Figure 20: Performance of $\gamma$ per-parameter methods. By default, we use $\gamma$ per layer and if specified, we use $\gamma$ per parameter.

### K.5 Qualitative behavior of soft resets

In this section, we zoom-in in the *data-efficient* experiment on random-label MNIST. We use *Soft Reset Proximal ($\gamma$ per layer)* method with separate $\gamma$ for layer (different for each weight and for each bias) and run it for 20 tasks on random-label MNIST. In Figure 21 we show the online accuracy as we learn over this sequence of tasks. In Figure 22, we visualize the dynamics of parameters $\gamma$ for each layer. First of all, we see that $\gamma_t$ seems to accurately capture the task boundaries. Second, we see that the amount by which each $\gamma_t$ changes depends on the parameter type – weights versus biases, and it depends on the layer. The architecture in this setting starts form $linear$ and goes up to $linear4$, which represent the 4 MLP hidden layers with a last layer $linear4$.

### K.6 Impact of specific initialization

In this section, we study the impact of using specific initialization $\theta_0 \sim p_{init}(\theta)$ in $p_0(\theta)$ as discussed in Appendix D. Using the specific initialization in *Soft Resets* leads to fixing the mean of the $p_0(\theta)$ to be $\theta_0$, see (24). This, in turn, leads to the predictive distribution (25). In case when we are not using specific initialization $\theta_0$, the mean of $p_0(\theta)$ is 0 and the predictive distribution is given by (23). To understand the impact of this design decision, we conduct an experiment on random label MNIST with *Soft Reset*, where we either use the specific initialization or not. For each of the variants, we

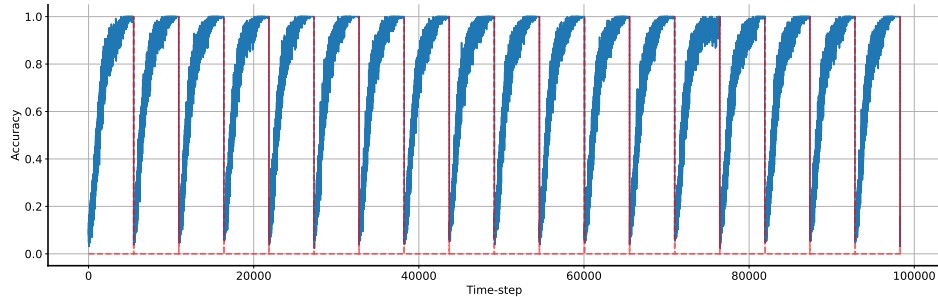

Figure 21: Visualization of accuracy when trained on *data efficient* random-label MNIST task. The dashed red lines correspond to a task boundary.

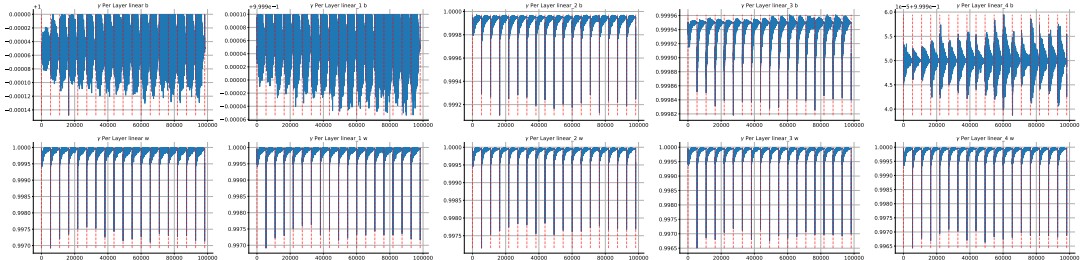

Figure 22: Visualization of $\gamma$ and task boundaries on *data-efficient* Random-label MNIST.

do a hyperparameters sweep. The results are given in Figure 23. We see that both variants perform similarly.

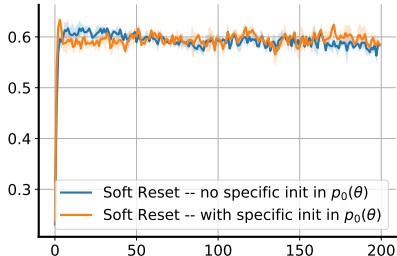

Figure 23: Impact of specific initialization $\theta_0$ as a mean of $p_0(\theta)$ in *Soft Resets*. The x-axis represents task id. The y-axis represents the average task accuracy with standard deviation computed over 3 random seeds. The task is random label MNIST – data efficient.

## L   Toy illustrative example for SGD underperformance in the non-stationary regime

**Illustrative example of SGD on a non-stationary stream.**   We consider a toy problem of tracking a changing mean value. Let the observations in the stream $\mathcal{S}_t$ follow $y_t = \mu_t + \sigma\epsilon$, where $\epsilon \sim \mathcal{N}(0,1)$, $\sigma = 0.01$. Every 50 timesteps the mean $\mu_t$ switches from $-2$ to $2$. We fit a 3-layer MLP with layer sizes $(10, 5, 1)$ and ReLU activations, using SGD with two different choices for the learning rate: $\alpha = 0.05$ and $\alpha = 0.15$. Moreover, given that we know when a switch of the mean happens, we reset (or not reset) all the parameters at every switch as we run SGD. Only during the reset, we use different learning rate $\beta = 0.05$ or $\beta = 0.15$. Using higher learning rate during reset allows SGD to learn faster from new data. We also ran SGD with $\alpha = 0.05$ and $\beta = 0.15$, where the higher learning rate is used during task switch but we do not reset the parameters. We found that it performed the same as SGD with $\alpha = 0.05$, which highlights the benefit of reset.

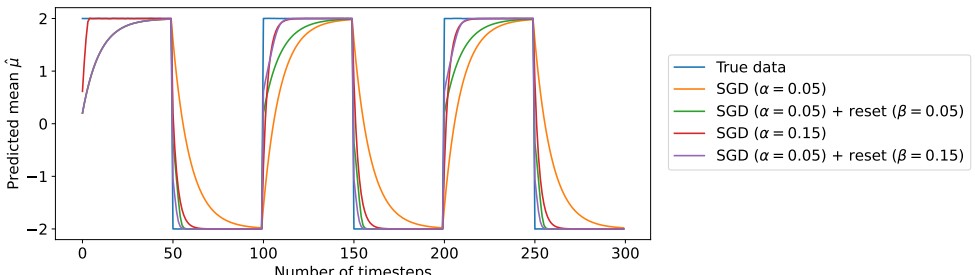

Figure 24: Non-stationary mean tracking with SGD.

We report the predicted mean $\hat{\mu}_t$ for all SGD variants in Figure 24. We see that after the first switch of the mean, the SGD without reset takes more time to learn the new mean compared to the version with parameters reset. Increasing the learning rate speeds up the adaptation to new data, but it still remains slower during the mean change from $2$ to $-2$ compared to the version that resets parameters. This example highlights that resets could be highly beneficial for improving the performance of SGD which could be slowed down by the implicit regularization towards the previous parameters $\theta_t$ and the impact of the regularization strength induced by the learning rate.

## M   Using arbitrary drift models

The approach described in section 3.5 provides a general strategy of incorporating arbitrary Gaussian drift models $p(\theta|\theta_t; \psi_t) = \mathcal{N}(\theta; f(\theta_t; \psi_t); g^2(\theta_t; \psi_t))$ which induces proximal optimization problem

$$\theta_{t+1} = \arg\min_\theta \widehat{\mathcal{L}}_{t+1}(\theta) + \frac{1}{2g(\theta_t; \psi_t)}||\theta - f(\theta_t; \psi_t)||^2 \tag{53}$$

The choice of $f(\theta_t; \psi_t)$ and $g(\theta_t; \psi_t)$ affects the behavior of the estimate $\theta_{t+1}$ from (53) and ultimately depends on the problem in hand. The objective function of the form (53) was studied in context of online convex optimization in [21],[29], where the underlying algorithms estimated the *deterministic* drift model online. These works demonstrated improved regret bounds depending on model estimation errors. This approach could also be used together with a Bayesian Neural Network (BNN).

