# OpenReview forum: "Non-Stationary Learning of Neural Networks with Automatic Soft Parameter Reset"
_NeurIPS.cc/2024/Conference — NeurIPS 2024 poster_

### Official Review · Reviewer_DfC5 · 2024-07-08

**Soundness:** 2
**Presentation:** 3
**Contribution:** 2
**Rating:** 5
**Confidence:** 3

**Summary:**

The paper focuses on non-stationary learning of neural networks. The paper proposes a method that automatically adjusts a parameter that influences the stochastic gradient descent (SGD) update to account for non-stationarity.

**Strengths:**

The problem tackled is relevant and the related work is clearly discussed. The method is well-interpreted and relevant for the community. The paper is also very well presented.

**Weaknesses:**

In my view, the paper has some weaknesses, especially regarding the experimental validation:
- The experimental validation does not clearly validate the benefits of the method, possibly due to lack of clarity. In Figure 3, it is not clear to me what method this paper proposes. I believe the paper should propose a method and validate it, instead of comparing all its possible variants. Nevertheless, I can see that most (if not all) of such variants are outperforming the chosen baselines. However, only two of the baselines referred in the related work were chosen. The chosen benchmark problems are also not motivated to me.
- The reinforcement learning part is not clear to me. Is there an explanation for why, on-policy, the method is outperformed by the simple baseline? Is not on-policy the most non-stationary setting? Why? And why does the proposed method stop earlier than the competitors in the top-right figure?
- It is not clear to me what the authors intend to show with Figure 2, and the perfect soft-resets regime. It would be nice to have more explanation here, and possibly move this validation to the end of the experimental section, as appearing before the main validation of the method is confusing.

Finally, even though the method is well motivated, it is not theoretically analyzed. We have no theoretical proof that the method will outperform traditional SGD in terms of efficiency, or preventing plasicity loss and catastrophic forgetting, or other dimension.

Minor: in page 6, Equation (15), what is the parameter $\tilde{\lambda}$? Was it introduced before?

**Questions:**

I have the following questions:
- Could the authors make some clarifications with respect to the experimental validation? Specifically, on the choice of baselines, problems, which method or methods the authors in fact propose to the community.
- Can the authors also clarify my concern regarding the reinforcement learning setting and the performance in on-policy settings?
- Can the authors clarify what is the intended take-away from the analysis in Figure 2?
- Can the authors clarify the Monte-Carlo samples in Equation (7)? Specifically, what is sampled, for instance in the reinforcement learning setting, and if the sampling requires access to a free simulator of the environment.

**Limitations:**

Yes.

---

> ### Author Rebuttal · Authors · 2024-08-06
>
> We thank Reviewer DfC5 for their feedback. Please find our response below.
>
> > Figure 3, hard to read
>
> Thanks to your feedback, we will modify the way we present the results, using the extended page limit for the camera-ready version. First, we will present **Soft Reset** method and compare it to baselines. Second, we will present other variants of the method and compare it to the main method.
>
> > baselines and benchmarks motivation
>
> The benchmarks are motivated by Plasticity Loss literature [1-4]. In this setting, Neural Networks lose ability to learn (lose plasticity) when exposed to certain forms of non-stationarity. Random label MNIST and CIFAR10 benchmarks are most common ones. Moreover, in this setting, resetting is highly beneficial (see [1]).
>
> We used 3 baselines, Shrink&Perturb [5], L2 Init [1] and Hard Resets. The Shrink&Perturb[5] is used to prevent plasticity loss by perturbing parameters according to $\theta_{t+1}=\lambda\theta_t+\sigma \epsilon$ where $\epsilon\sim\mathcal{N}(0;I)$. L2 Init approach adds $||\theta - \theta_0||^2$ to the loss and was shown to outperform many other approaches in preventing plasticity loss. Both baselines are highly related to our drift model and discussed in Related Work. We hope this clarifies the motivations.
>
> > RL is not clear...
>
> Please see our response to Reviewer Hmj6.
>
> > Why in on-policy, simple method performs better than ours, given that it is more non-stationary setting?
>
> In RL, the non-stationary rises from changing input data distribution (since we change the policy) as well as from changing learning targets (when learning value functions). In the off-policy RL, we expect a high impact of changing learning target non-stationarity. It was observed [4,6-7], that in this setting we can exhibit the loss of plasticity. The effect becomes more present as we increase the replay ratio (increase the number of updates per collected data). Under high replay ratio since we do many updates on replay buffer, we could overfit to the replay buffer and when we start collecting new data, the local minima of NN can switch drastically. This is the regime which motivated our method (see Figure 1 and beginning of Section 3). In this setting, hard resets [4] are effective but our method shows to be even more effective.
>
> When we operate in on-policy setting, the situation described above is less likely to happen. If data distribution changes quickly, then we are less likely to overfit to it due to the noise in the update. Using the language from Figure 1, the uncertainty of the Bayesian posterior will not be shrinking if the data changes quickly. This, however, depends on the environment. The Hopper environment (Figure 4, bottom) is a relatively simple task to solve, meaning that the agent will progress fast and the corresponding input data distribution will change fast. The Humanoid environment (Figure 4, top) is a much more challenging environment, and the simple agent might struggle initially. This implies that even on-policy algorithm can see a lot of similar data and start to overfit to it, putting it closer to the off-policy setting. This explains why our method is more effective in Humanoid rather than Hopper environment when relpay ratio is 1.
>
> > why stops earlier?
>
> The experiment was not fully finished at the time of submission. See attached pdf for final version of the Figure.
>
> > Figure 2 ?
>
> In Figure 2, we demonstrate that drift model eq.4 coupled with update rule eq.16 is a good strategy of resetting parameters when task boundaries are known. For appropriately chosen $\gamma \neq 0$, we can achieve significantly better performance than hard reset. In Figure 2, left, we use constant l.r. $\alpha_t$ whereas in Figure 2,right, we use $\alpha_t(\gamma_t)$ from eq.17, which is more beneficial. We will add more clarifications in the text about the nature of this experiment.
>
> > Theoretical analysis
>
> Unfortunately, the phenomenon of plasticity loss in Neural Networks is not theoretically analyzed (there is no model for that), it is an empirical phenomenon [1-4]. We would require this theoretical framework for plasticity loss to be developed in order to analyze our method. Our future plans involve analyzing the method in the context of online convex optimization.
>
> > MC samples in eq.7?
>
> MC samples are not coming from a simulator but from a Gaussian distribution eq.5 induced by the drift model eq.4. MC samples are needed to approximate integral in eq.7. The MC samples correspond to NN parameters.
>
> Overall, we hope our answer clarifies the points which you raised during your review. If we have addressed your questions, we would be  grateful if you would consider increasing your score. We would be happy to answer any further questions you might have.
>
> **References**:
>
> [1] Maintaining Plasticity in Continual Learning via Regenerative Regularization, Saurabh Kumar, Henrik Marklund, Benjamin Van Roy, 2023
>
> [2] Understanding plasticity in neural networks, Clare Lyle, Zeyu Zheng, Evgenii Nikishin, Bernardo Avila Pires, Razvan Pascanu, Will Dabney, 2023
>
> [3] Disentangling the Causes of Plasticity Loss in Neural Networks, Clare Lyle, Zeyu Zheng, Khimya Khetarpal, Hado van Hasselt, Razvan Pascanu, James Martens, Will Dabney, 2024
>
> [4] Understanding and Preventing Capacity Loss in Reinforcement Learning, Clare Lyle, Mark Rowland, Will Dabney, 2022
>
> [5] On Warm-Starting Neural Network Training, Jordan T. Ash, Ryan P. Adams, 2020
>
> [6] The Primacy Bias in Deep Reinforcement Learning, Evgenii Nikishin, Max Schwarzer, Pierluca D'Oro, Pierre-Luc Bacon, Aaron Courville, 2022
>
> [7] The Dormant Neuron Phenomenon in Deep Reinforcement Learning, Ghada Sokar, Rishabh Agarwal, Pablo Samuel Castro, Utku Evci, 2023

---

> > ### Comment · Reviewer_DfC5 · 2024-08-12
> > **Response to authors' rebuttal**
> >
> > I thank the authors for the response.
> >
> > I had some points clarified. However, I am inclined to maintain my score, since the experimental validation is somehow confusing. Regarding the RL results, I can not see clear benefits from the approach. Regarding Figure 3, the authors replied that their method is Soft Reset, but I see it being outperformed by several of its variants. Then why do the authors choose as their proposed method Soft Reset and not one of the variants that outperform it?
> >
> > Thank you.

---

> > > ### Author Response · Authors · 2024-08-12
> > > **Response to a response**
> > >
> > > Dear Reviewer DfC5, thank you for your response.
> > >
> > > > Regarding Figure 3, the authors replied that their method is Soft Reset, but I see it being outperformed by several of its variants. Then why do the authors choose as their proposed method Soft Reset and not one of the variants that outperform it?
> > >
> > > It is true that it is outperformed by other variants. Soft Reset is the cheapest method among all the other variants -- it requires only one update on $\gamma_t$ and one update on parameters $\theta_{t+1}$. All the other methods -- Soft Reset more compute, Soft Reset proximal, Bayesian Soft Reset proximal, require significantly more compute. Figure 3 suggests that we can in fact leverage more compute to achieve better empirical performance.
> > >
> > > This is why we chose to present Soft Reset as the main method because it is relatively cheap and already achieves good performance compared to external baselines. However, when more compute is available, other methods could also be used to further improve performance.
> > >
> > > Hope this clarifies the confusion.

---

### Official Review · Reviewer_4koq · 2024-07-13

**Soundness:** 2
**Presentation:** 1
**Contribution:** 2
**Rating:** 4
**Confidence:** 3

**Summary:**

- The authors study a learning algorithm that can handle the non-stationarity of the data distribution.
- They propose a parameter drift model based on the Ornstein-Uhlenbeck process, which models a form of “soft parameter reset” adaptive to the data stream. The drift model has an adaptive parameter $\gamma_t$ which a gradient-based optimizer can learn online.
- They illustrate the update rule of the main parameters of a neural network incorporating the learned drift model. The update rule is first proposed under the Bayesian neural network framework and then later adapted to a non-Bayesian neural network.
- They numerically corroborate the efficacy of their method on plasticity benchmarks and reinforcement learning experiments.

**Strengths:**

- S1. They propose a novel way to learn online the amount of resetting the parameters.
- S2. The method is general enough to be applied broadly to continual learning problems and reinforcement learning problems.

**Weaknesses:**

- W1. Modeling parameter drift is not well-motivated.
    - The learnable drift model $p(\theta_{t+1} \mid \theta_t, \gamma_t)$ is the main contribution of this work. In my opinion, however, it is unclear why we should care about the drift of “currently learning” neural network parameters.
    - According to the beginning of Section 3, the local minima of the objective function may change over time. This is of course acceptable. However, I cannot understand what is relevant between the change of the local optima $\theta^{\star}_t$ and the current parameter $\theta_t$. To be more specific, the two sentences in lines 94-96 are not connected well.
- W2. The paper is NOT self-contained overall.
    - The authors claim that they chose the Ornstein-Uhlenbeck (OU) process as a drift model. However, it is unclear how a continuous stochastic differential equation (OU process) can be converted to a discrete Markovian chain defined in Equation (4). The paper does not even introduce the actual form of the OU process. Moreover, it is not very clear why the authors chose the particular model defined in Equation (4) because it is not known (at least in the paper) that the model is a unique option to choose.
    - There are a lot of equations whose derivation is omitted. Let me list them: Equations (9), (10), (11), (14), (15). In particular, I am suspicious of both the validity and usefulness of Equation (10).
    - For these reasons, the paper is not so easy to follow.
- W3. The soft reset method seems computationally heavy.
    - I am worried about the computational cost. Although it is good to learn the forgetting parameter $\gamma_t$ online, it increases the computational cost almost twice. It might be more than just twice because learning the $\gamma_t$ requires Monte-Carlo (MC) sampling.
    - In addition, there seem too many hyperparameters to tune.
- W4. Minor comments on typos/mistakes
    - Do not capitalize the word “neural network”. (e.g., line 35-36)
    - The term “plasticity loss” may be read as a type of loss function for some people (new to this field). I recommend using the term “loss of plasticity”.
    - Line 84: “non-stationary” → “non-stationarity”
    - Line 109: “mportant” → “important”
    - Line 127: “wrt” → “with respect to”. Do not use abbreviations.
    - Line 137: “property” → “properties”
    - Line 154: what is “s” next to a parenthesis?
    - Line 168: “$\theta = (\theta_i, \ldots, \theta_D)$” → “$\theta = (\theta_1, \ldots, \theta_D)$”
    - Line 169: “$\mathbb{R}^D$” is a more standard notation than “$\mathcal{R}^D$”.
    - Line 222: In “$\lambda\_i = \hat{\lambda}\_i \sigma^2_{t,i}$", I think $\hat{\lambda}$ must not have a subscript.
    - Line 225: “sinc” → “since”
    - Equation (14): What is “$\tilde{\mathcal{F}}$”? (Partial) gradient of “$\mathcal{F}$”?
    - Line 232: What does it mean by “we assume that $\mu_0 = \theta_0, \sigma_0^2.$”?
    - Line 236: “$\theta_{t,i}(\gamma_{t,i}) = \theta_t$” → “$\theta_{t,i}(\gamma_{t,i}) = \theta_{t,i}$”
    - Line 240: “linearisng” → “linearizing” or “linearising”
    - Line 292: duplicate “See”
    - Line 354: “modelling” → “modeling”
    - Lines 492-493: I guess “*data efficient*” and “*memorization*” are swapped.

**Questions:**

Q1. As far as I know, the Hard Reset method typically resamples the model parameter every time it resets the parameter. With this in mind, what if we slightly modify the drift model as: $p(\theta \mid \theta_t, \gamma_t) = \mathcal{N} (\theta; \gamma_t \theta_t + (1-\gamma_t) \theta’_0 ; (1-\gamma_t^2)\sigma_0^2)$, where we re-sample $\theta’_0 \sim p_0(\theta_0)$ every time $t$?

**Limitations:**

The paper discusses its limitations in the experiment section. I think it might be better to mention them explicitly in the conclusion section.

---

> ### Author Rebuttal · Authors · 2024-08-06
>
> We thank reviewer 4koq for their response. Please find our detailed answer below.
>
> > why drift model...
>
> In Section 3 and in Appendix D, we presented the reasons to use a drift model together with learning NN parameters. Figure 1 illustrates the high-level intuition in case of online Bayesian estimation. Using SGD language, assuming that the data is stationary up to the time $T$, SGD estimate $\theta_{T}$ would tend towards a local optima $\theta^*$. If data stays stationary at time $T+1$, $\theta_{T+1}$ will move closer to the $\theta^*$. However, if at time $T+1$, the data distribution changes and so is the set of local optima, SGD might struggle to move towards this set starting from $\theta_T$. Drift model allows the learning algorithm to make larger moves towards this new set of local optima. Such idea was also used in the context of online convex optimization, see [2-3]. The form in eq.4 encourages parameters to return back towards the initialization over time by shrinking the mean and increasing the variance (l.r. in SGD), allowing to make bigger steps towards new local minima.
>
> We will change the wording of the beginning of Section 3 to better reflect this explanation.
>
> > form of OU process
>
> OU process defines a SDE that can be solved explicitly and written as a time-continuous Gaussian Markov process with transition density $p(x_t|x_s)=\mathcal{N}(x_s e^{-(t-s)},(1-e^{-2(t-s)})\sigma_0^2I)$ for any pair of times  $t>s$. Based on this, as a drift model for the parameters $\theta_t$ (so $\theta_t$ is the state $x_t$) we use the conditional density $p(\theta_{t+1}|\theta_t)=\mathcal{N}(\theta_t\gamma_t,(1-\gamma_t^2)\sigma_0^2I)$ where $\gamma_t=e^{-\delta_t}$ and $\delta_t\geq0$ corresponds to the learnable discretization time step. In other words, by learning $\gamma_t$ online we equivalently learn the amount of a continuous “time shift” $\delta_t$ between two consecutive states in the OU process. This essentially models parameter drift since e.g. if $\gamma_t=1$, then $\delta_t=0$ and there is no “time shift" implying $\theta_{t+1}=\theta_t$.
>
> > ...why eq.4
>
> In Section 3, we argued for choosing eq.4 because it pushes the learning towards the initialization as well as towards previous parameters, allows to use gradient-based methods to estimate drift parameters and keeps positive finite variance (assuming we learn over the infinite amount of time) which avoids degenerate cases of $0$ or $\infty$ variance. Moreover, it couples the mean and the variance via $\gamma$, making it easier to learn $\gamma$ and less likely to overfit. Other potential choices for the drift models:
>
> * $\theta_{t+1}=\gamma\theta_t+(1-\gamma)\mu_0+\beta\epsilon$, with $\epsilon\sim\mathcal{N}(0,I)$
>
>   * Fixed $\beta$ was explored in our experiment in Figure 2, left, where we used constant l.r. $\alpha_t$ for parameters update. We found that it performed worse than using rescaled l.r. $\alpha(\gamma_t)$ from eq.17 which is derived using our model, see Figure 2, right. For $\mu_0=0$, we recover Shrink&Perturb [4]
>   * Learning $\gamma$ and $\beta$ will likely overfit
>
> * Mixture $p(\theta_{t+1}|\theta_t,\gamma_t)=\gamma_tp(\theta_{t+1}|\theta_t)+(1-\gamma_t)p_0(\theta_{t+1})$, where $p(\theta_{t+1}|\theta_t)=\mathcal{N}(\theta_t;\sigma^2)(\theta_{t+1})$ and $p_0(\theta_{t+1})=\mathcal{N}(\mu_0;\sigma_0^2)(\theta_{t+1})$, which is a Gaussian version of Spike&Slab [5]. This encourages **hard** resets instead of **soft** ones. Moreover, using mixtures is problematic since the KL in eq.11 cannot be computed exactly and needs to be approximated.
>
> We will add the discussion of drift model choices in the appendix.
>
> > derivations...
>
> Please see our reply to Reviewer 6GY1 about the eq.9 and eq.11. Eq.10 is obtained from eq.9 by finding fixed points. In eq.10, we have a typo, the term $\lambda\gamma_t^0$ in the denominator should be replaced by $\lambda$. Eq.14 is a gradient descent rule applied to eq.11(eq.12) where $\tilde{F}$ is actually a derivative of $F$ wrt $\mu_t$ and $\sigma_t$. Eq.15 is Maximum a-posteriori (MAP) update for $\theta$ from the posterior $p(y_{t+1}|x_{t+1},\theta)q^{t+1}{t}(\theta |\gamma_t)$ and where we use temperature in the prior (see [3]).
>
> We will add all the derivations and explanations in the appendix.
>
> > $\theta'$ every time
>
> While a possible modification for the drift model, it could be problematic to use, since resampling $\theta_{0}$ has higher variance than OU process. The variance of this model is $\gamma_t^2\sigma^2_t+(1-\gamma_t)^2\sigma^2_0+(1-\gamma^2_t)\sigma^2_0$, which equals to $2\sigma^2_0$ for $\gamma_t=0$ which is 2x larger than the variance of the initialization. This implies that such a model might inflate the variance over time since e.g. if $\gamma_t=0.5$ for all time steps $t$, then $\sigma_{t+1}^2=\gamma_t^2 \sigma^2_t+2(1-\gamma_t)\sigma^2_0=0.5^2\sigma^2_t+\sigma^2_0$ which grows with the iterations.
>
> Thus, we believe that our current drift model more accurately implements the mechanism of parameter resets since it always  brings us back to the exact initialization distribution.
>
> We hope we have been able to address your concerns and we are happy to answer further questions on the above subjects. If we were able to address your concerns, we would be grateful if you would consider increasing your review score.
>
> **References**:
>
> [1] Dynamical Models and Tracking Regret in Online Convex Programming, Eric C. Hall, Rebecca M. Willett, 2013
>
> [2] Adaptive Gradient-Based Meta-Learning Methods, Mikhail Khodak, Maria-Florina Balcan, Ameet Talwalkar, 2019
>
> [3] How Good is the Bayes Posterior in Deep Neural Networks Really?, Florian Wenzel, Kevin Roth, Bastiaan S. Veeling, Jakub Świątkowski, Linh Tran, Stephan Mandt, Jasper Snoek, Tim Salimans, Rodolphe Jenatton, Sebastian Nowozin, 2020
>
> [4] On Warm-Starting Neural Network Training, Jordan T. Ash, Ryan P. Adams, 2020
>
> [5] Spike and slab variable selection: Frequentist and Bayesian strategies, Hemant Ishwaran, J. Sunil Rao, 2005

---

> ### Comment · Reviewer_4koq · 2024-08-10
> **The response is not satisfactory yet**
>
> I have read the author’s rebuttal, but I still have remaining concerns and questions. Let me leave my comments/questions about the author’s responses one at a time.
>
> 1. I don’t think this answer really responds to my question in W1. Here, I am not asking about the motivation of the drift model (although I did it in W2). My question is: why do we care about the conditional distribution of $\theta_{t+1}$, a consequence of a single step of training from $\theta_t$, even though we want to estimate or utilize the dynamics of the moving local optima $\theta^{\ast}\_t \mapsto \theta^{\ast}\_{t+1}$?
>     - After pondering this issue and staring at Figure 1, I suddenly realized that the drift model is actually modeling the dynamics of $\theta^{\ast}\_{t+1}$ given $\theta_t$. So in my understanding, the $\theta_{t+1}$ in “$p(\theta_{t+1} \mid \theta_t, \gamma_t)$” actually means $\theta^{\ast}_{t+1}$, the local optima at time $t+1$, rather than the parameter we will have by updating the parameter from $\theta_t$ using a single step of learning algorithm. (Or, at least, we want to find a local optimum $\theta^{\ast}\_{t+1}$ close to the initialization & $\theta_t$.) Is my understanding correct? If it is, I think this should have been explained in the paper in more detail; also, the term “parameter drift” is a bit misleading in this sense.
>     - By the way, I guess the citation ([2-3]) has a typo, right? It seems that it should include [1].
> 2. Thank you for the explanation of the relationship between the OU process, the Gaussian Markov process, and the choice of the drift model. Will the authors add this explanation to their paper? Or do they just ignore it?
> 3. Thank you for the discussion on the other options for drift models and for considering putting the discussion to the paper.
> 4. Thank you for derivations of the equations in more detail. However, I think the derivations could be more kind than the current explanation.
>     - Eq 9: Please state to which parameter the linearization is taken (e.g., By linearizing log… around $\theta=\mu_t$ (cf. Line 198))
>     - Eq. 11 & 12: Is these are just the usual objective functions in BNN training?
> 5. Thank you for the interesting response based on the drift model and the variance. But what if we change the **variance scheduling**: what if we suitably change the drift model into $p(\theta \mid \theta_t, \gamma_t) = \mathcal{N} (\theta; \gamma_t \theta_t + (1-\gamma_t) \theta’_0(t); 2\gamma_t(1-\gamma_t)\sigma_0^2)$ where $\theta’_0(t) \sim p_0(\theta_0)$? I guess this model does not suffer from the same problem of variance inflation: If $\sigma_t = \sigma_0$, then ${\rm Var}(\theta) = \gamma_t^2 \sigma_t^2 + (1-\gamma_t)^2 \sigma_0^2 + 2\gamma_t (1-\gamma_t) \sigma_0^2 = \\{\gamma_t^2 + (1-\gamma_t)^2 + 2\gamma_t (1-\gamma_t)\\}\sigma_0^2 = \sigma_0^2$...! The motivation behind this is the concern about the dependency on the particular initialization “$\theta_0$” which seems to be fixed throughout the training: “What if the initialization distribution is only the matter?” Please let me know if there are other problems in this drift model, or I would be very happy if you could test this drift model empirically (but I understand if it is impossible due to the time limit)!
>
> Also, there is a question that is not answered yet. Let me bring it here:
>
> - Line 232: What does it mean by “we assume that $\mu_0 = \theta_0, \sigma_0^2.$”?
>
> Although I appreciate the time and effort invested in the rebuttal and am happy with the author's general response, overall, I feel like the author’s response is not really satisfying yet. Even though the authors requested to reconsider my assessment, I cannot do so before I get a satisfying further response. If the further response is not satisfactory as well, sadly and unfortunately, I am **ready** to decrease my score to 3 (but **not yet**) because, in my view, there is still a huge room for improvement in writing and presentation.

---

> ### Author Response · Authors · 2024-08-12
> **Response to a response: Part 1**
>
> Dear 4koq, thank you for your feedback and please find our answer to your concerns below.
>
> > I don’t think this answer really responds to my question in W1. Here, I am not asking about the motivation of the drift model (although I did it in W2). My question is: why do we care about the conditional distribution of
> Θt+1 , a consequence of a single step of training from  θt , even though we want to estimate or utilize the dynamics of the moving local optima  θt∗↦θt+1∗?
> After pondering this issue and staring at Figure 1, I suddenly realized that the drift model is actually modeling the dynamics of  θt+1∗  given  θt.. So in my understanding, the  θt+1  in “p(θt+1∣θt,γt)” actually means  θt+1∗ , the local optima at time  t+1, rather than the parameter we will have by updating the parameter from θt  using a single step of learning algorithm. (Or, at least, we want to find a local optimum θt+1∗ close to the initialization & θt .) Is my understanding correct? If it is, I think this should have been explained in the paper in more detail; also, the term “parameter drift” is a bit misleading in this sense.
>
> We apologize for the confusion, but under the Bayesian perspective depicted in Figure 1 $\theta^*_{t}$ and $\theta^*_{t+1}$ denote **fixed/deterministic** values and therefore are not random variables, while the corresponding random variables which are assigned probability distributions are $\theta_{t}$ and $\theta_{t+1}$. More specifically, we learn the dynamical model  distribution $p(\theta_{t+1}|\theta_{t},\gamma_t)$, by learning $\gamma_t$, so that the specific value $\theta^*_{t+1}$ can become more likely or “explainable” under the distribution $p(\theta_{t+1}|\theta_{t},\gamma_t)$. In other words we hope that $p(\theta_{t+1}|\theta_{t},\gamma_t)$ will place a considerable probability mass around the fixed value $\theta^*_{t+1}$.   We are going to update the paper  to make the above clear and remove the confusion.
>
> To further explain here this in Figure 1, consider the stationary case depicted in Figure,1a. There the posterior concentrates around an optimal value (which is fixed value $\theta^*$), in other words $q_t(\theta)$ will gradually converge to a delta mass around the value $\theta^*$.  In a non-stationary case, the optimal value can suddenly change over time and e.g. from the value $\theta^*_{t}$ at time $t$ can change to $\theta^*_{t+1}$ . Without a dynamical model  (Figure 1,b)  the new posterior $q_{t+1}(\theta)$ after observing the new data at time $t+1$ has a small radius/variance (blue dashed circle) and it cannot concentrate fast enough towards the new optimum.  The use of  the dynamical or drift model (Figure 1,c) introduces an “intermediate step” that constructs an “intermediate prior distribution”  $p_t(\theta) = \int q_t(\theta) p(\theta | \theta_t, \gamma_t) d \theta_t$ which can have increased variance and shifted mean  (green dashed circle). Then  once we fully incorporate the  new data point at time $t+1$ the updated posterior $q_{t+1}(\theta) \propto p(y_{t+1}|\theta) p_t(\theta)$ can better concentrate around $\theta_{t+1}^*$. We will update the paper to clarify the above.
>
> > citation [2-3] is a typo
>
> Thank you, it should be [1-2].
>
> > Q2
>
> Yes we will add this to the paper (we forgot to explicitly write it).
>
> > Q3
>
> Thank you.
>
> > Eq.11&12 -- usual BNN objective?
>
> Yes, except for the temperature defined per-parameter. In BNN, temperature $\lambda$ is either 1, or fixed to a constant for all the parameters.
>
> > Line 232: What does it mean by “we assume that $\mu_0=\theta_0,\sigma^2_0.$?
>
> It means that the mean of the prior $p_0(\theta)=\mathcal{N}(\theta;\mu_0,\sigma^2)$ is fixed to $\theta_0$, i.e. the initialization and $\sigma^2_0$ is a constant. Normally NNs are initialized from $p_0(\theta)=\mathcal{N}(\theta;0,\sigma^2_0)$, but we make the prior to be initialization-dependent. We will rewrite this sentence to explicitly highlight this.
>
> We hope your answer clarifies the questions and concerns you have raised, and we thank you for raising interesting points for discussion!

---

> > ### Author Response · Authors · 2024-08-12
> > **Part 2**
> >
> > > Eq.9, linearisation
> >
> > As we stated above in the rebuttal, the response to Reviewer 6GY1 (due to space constraints) contains the partial derivation. Linearisation is $\log p(y_{t+1}|x_{t+1},\theta)\sim\log p(y_{t+1}|x_{t+1},\mu_t)-g_{t+1}^T(\theta-\mu_t)$, where $g_{t+1}=\nabla_{\theta} -\log p(y_{t+1}|x_{t+1},\theta=\mu_t)$. Then, $p(y_{t+1}|x_{t+1},\theta) \sim p(y_{t+1}| x_{t+1},\mu_t)\exp^{-g_{t+1}^T\mu_t}\exp^{g_{t+1}^T\theta}$. Since, $q_t^{t+1}(\theta | S_{t}, \Gamma_{t-1}, \gamma_t)$ is Gaussian, we compute the integral eq.6 in a closed form and keep only the terms depending on $\gamma_t$. We provide derivation for a scalar case here and full derivation in the appendix. In eq.6, we have
> > $\log\int p(y_{t+1}|x_{t+1},\theta)\exp^{-\frac{(\theta -\mu_t(\gamma_t))^2}{2\sigma^2(\gamma_t)}}d\theta =\log \frac{1}{\sqrt{2\pi\sigma^2(\gamma_t)}}\int p(y_{t+1}|x_{t+1},\mu_t)\exp^{-g_{t+1}(\theta-\mu_t)} \exp^{-\frac{(\theta-\mu_t(\gamma_t))^2}{2\sigma^2(\gamma_t)}}d\theta$
> >
> > which becomes
> >
> > $\log p(y_{t+1}|x_{t+1},\mu_t)+\log \int \frac{1}{\sqrt{2\pi\sigma^2(\gamma_t)}}\exp^{-g_{t+1}(\theta-\mu_t)}\exp^{-\frac{(\theta-\mu_t(\gamma_t))^2}{2\sigma^2(\gamma_t)}}d\theta$
> >
> > Consider only the exponent term in the integral
> >
> > $\frac{-1}{2\sigma^2(\gamma_t)}(2\sigma^2(\gamma_t)g_{t+1}(\theta-\mu_t)+\theta^2-2\theta \mu_t(\gamma_t)+\mu_t(\gamma_t)^2)$
> >
> > then
> >
> > $\frac{-1}{2\sigma^2(\gamma_t)}\left(\theta^2-2\theta \left[\mu_t(\gamma_t)-\sigma^2(\gamma_t)g_{t+1}\right]+\mu_t(\gamma_t)^2-2\sigma^2(\gamma_t)g_{t+1}\mu_t\right)$
> >
> > then
> >
> > $\frac{-1}{2\sigma^2(\gamma_t)}\left[ \left(\theta-(\mu_t(\gamma_t)-\sigma^2(\gamma_t)g_{t+1}) \right)^2+2\mu_t(\gamma_t) \sigma^2(\gamma_t)g_{t+1}-\sigma^4(\gamma_t)g_{t+1}^2-2\sigma^2(\gamma_t)g_{t+1}\mu_t)\right]$
> >
> > then
> >
> > $\frac{-1}{2\sigma^2(\gamma_t)} \left(\theta-(\mu_t(\gamma_t)-\sigma^2(\gamma_t)g_{t+1})\right)^2-\mu_t(\gamma_t)g_{t+1}+0.5\sigma^2(\gamma_t)g^2_{t+1}+g_{t+1}\mu_t$
> >
> > Now, get back to the integral
> >
> > $\log\int\frac{1}{\sqrt(2\pi\sigma^2(\gamma_t)}\exp^{\frac{-1}{2\sigma^2(\gamma_t)}\left(\theta-(\mu_t(\gamma_t)-\sigma^2(\gamma_t)g_{t+1}) \right)^2-\mu_t(\gamma_t)g_{t+1}+0.5\sigma^2(\gamma_t)g^2_{t+1}+g_{t+1}\mu_t}d\theta$
> >
> > Which equals to
> >
> > $1-\mu_t(\gamma_t)g_{t+1}+0.5 \sigma^2(\gamma_t)g^2_{t+1}+g_{t+1}\mu_t$
> >
> > We keep only the terms which depend on $\gamma_t$ and get
> >
> > $G(\gamma_t)=-\mu_t(\gamma_t)g_{t+1}+0.5 \sigma^2(\gamma_t)g^2_{t+1} $
> >
> > We want to maximize $G(\gamma_t)$ or minimize $F(\gamma_t)=-G(\gamma_t)$ which is exactly the loss we defined in eq.9.
> >
> > We will add the full derivation in the appendix.
> >
> > > Different drift model
> >
> > Assuming that $\theta'\sim\mathcal{N}(\theta;\mu_0;\sigma^2_0)$, the model you wrote will have the mean $\gamma_t\theta_t+(1-\gamma_t)\mu_0$ and the variance $\gamma^2_t \sigma^2_t+(1-\gamma^2_t)\sigma^2_0$, which are the same as for our OU model, see eq.4 and Line 173 after eq.5, therefore they are mathematically equivalent.
> >
> > In order to discuss it in detail, let $\mu_0$ is the prior mean for eq.4 and $\mu'_0$ is the prior mean in your variant, i.e., $\theta'\sim\mathcal{N}(\theta;\mu'_0;\sigma^2_0)$. When we incorporate the Gaussian model drift in the update, we are only concerned with its mean and variance since these affect SGD updates (see eq.16).
> >
> > We have the following three cases:
> >
> > * $\mu_0=\mu_0'=0$, the models have the same mean equal to $0$. They are mathematically equivalent to the model $\theta_{t+1}=\gamma_t\theta_t+\sqrt{1-\gamma^2_t}\theta_0$, where $\theta_0\sim p_0(\theta)=\mathcal{N}(\theta;0; \sigma^2)$ -- the models are the same. The have the mean $\gamma_t\mu_t$ and the variance $\gamma^2_t\sigma^2_t+(1-\gamma^2_t)\sigma^2_0$.
> >
> > * $\mu_0=\mu'_0=\theta_0\neq0$, the models have the same mean which are the one specific initialization of NN parameters. They are mathematically equivalent with the mean $\gamma_t\mu_t+(1-\gamma_t)\theta_0$ and the same variance (as above)
> >
> > * $\mu_0=\theta_0$ and $\mu'_0 \neq \theta_0$. These two models have the same variance, $\gamma_t^2 \sigma^2_t + (1-\gamma^2_t)\sigma^2_0$, but have different means: $\gamma_t\theta_t+(1-\gamma_t)\theta_0$ versus $\gamma_t\theta_t+(1-\gamma_t)\mu'_0$.
> >
> > To implement variant 3, the mean $\mu'_0$ is fixed at the beginning of the training, because otherwise we would be under-counting the variance in the corresponding drift model. It is unclear how to interpret this model nor how it will affect performance.
> >
> > Thus, in order to answer your question about the impact of the particular initialization, we believe variant 1 is the right answer. We conducted an experiment of using Soft Reset variant 1 and 2 on random label MNIST (data efficient), where we swept over hyperparameters. We found that these variants had similar performance on this benchmark. We will add this experiment to the paper in the Appendix.
> >
> > We hope your answer clarifies the questions and concerns you have raised, and we thank you for raising interesting points for discussion!

---

> > > ### Comment · Reviewer_4koq · 2024-08-12
> > >
> > > I thank the authors for a couple of more detailed replies.
> > >
> > > 1. About original drift model: Thank you for resolving most of my confusion. Now I think I understand the concept of drift model. Still, I am concerned about that, in the author's explanation, the local optimum $\theta^{\ast}\_t$ seems a unique deterministic value/vector. However, in the era of overparameterized deep learning and because of its nonconvex nature, the $\theta^{\ast}\_t$ may not be unique; rather, there might be a set of (non-unique) local optima. Because of this, I thought the drift model is a kind of mechanism of choosing a proper/desired $\theta^{\ast}\_t$ among all local optima. Although I admit that $\theta^{\ast}\_t$ not a random object (but still it can be because of randomness in data sampling) and $\theta_t$ is an estimate of it (thus it is definitely a random object), considering the non-uniqueness of local optima, it seems vague and non-rigorous for me to say "getting closer to $\theta^{\ast}\_t$" or "concentrating aroung $\theta^{\ast}\_t$".
> > >
> > > 2. About linearization: I am already aware of the derivation explained for reviewer 6GY1. All I wanted is just a kind derivation in your next revised paper. I wanted to take part in improving your paper.
> > >
> > > 3. About variations of drift model: thank you for clearly answering my question. Personally, I was excited for suggesting a new point of view of your methodology, but it turns out it was somewhat meaningless (which is sad). Anyway, I am happy to understand the significance of the original formulation of the drift model.
> > >
> > > Thank you again for the detailed response. Nonetheless, in my opinion, it would be better to judge a completely rewritten manuscript with another review process. Therefore, I maintain my score.

---

> > > > ### Author Response · Authors · 2024-08-13
> > > > **Response**
> > > >
> > > > We thank the Reviewer 4koq for their response and for sparking interesting discussion! Please see our answer below.
> > > >
> > > > > I am concerned about that, in the author's explanation, the local optimum $\theta^{\ast}_t$ seems a unique deterministic value/vector. However, in the era of overparameterized deep learning and because of its nonconvex nature, the $\theta^{\ast}_t$ may not be unique; rather, there might be a set of (non-unique) local optima.
> > > >
> > > > Please note that when we discuss local optimas $\theta^*_{t}$ and $\theta^*_{t+1}$, we do not assume their uniqueness. Rather, the only assumption we make is that our algorithm may converge to some local optima $\theta^*_{t}$ (which is a member of a set of local optima), and after non-stationarity, a new set of local optima will appear, where one of the member is $\theta^*_{t+1}$. The reason why we use two specific points $\theta^*_{t}$ and $\theta^*_{t+1}$ in Figure 1, is to simplify the understanding and communicate the intuition behind the non-stationary behaviour and the impact of a drift model.
> > > >
> > > > In the paper, in Section 3.1, we mainly refer to a region of new local minima, please see, for example, Lines 110-112: "If the change is such that the region of new good local minima is far away from $\theta^*$, SGD may take considerably higher amount of time to move its parameters towards this region."
> > > >
> > > > Hope this clarifies the concerns.
> > > >
> > > > > About linearization: I am already aware of the derivation explained for reviewer 6GY1. All I wanted is just a kind derivation in your next revised paper. I wanted to take part in improving your paper.
> > > >
> > > > Thanks to the useful discussion with you and the concerns you have raised, we think that including the full derivation of the linearization in the appendix of the paper, will greatly reduce confusion shared by you and other reviewers. We thank you for your suggestions, we will add more context in the paper.

---

### Official Review · Reviewer_6GY1 · 2024-07-13

**Soundness:** 4
**Presentation:** 4
**Contribution:** 3
**Rating:** 7
**Confidence:** 3

**Summary:**

The paper proposes a method to effectively learn the neural network parameters in non-stationary environments. They propose a modified learning algorithm that adapts to non-stationarity through an Ornstein-Uhlenbeck process with an adaptive drift parameter. Drift parameter is used to track the non-stationarity in the data - specifically, adaptive drift pulls the NN parameters towards initial parameters when the data is not well-modelled by current parameters while on the other hand reducing the degree of regularization (towards initial parameters) when the data is relatively stationary. The paper introduces a Bayesian Soft-Reset algorithm in Bayesian Neural Network (BNN) framework and a General Soft-Reset algorithm in deterministic non-Bayesian framework to effectively learn in non-stationary environments. Empirically, authors demonstrate that their method performs effectively in non-stationary supervised and off-policy reinforcement learning settings.

**Strengths:**

To take into account the non-stationarity in the data - the methodology in the paper modifies the SGD algorithm by modifying the regularization strength and the regularization target of the SGD algorithm - a significant contribution of the work is to do this modification in a principled manner using a drift parameter that is updated online from the stream of data.

Overall, the paper is well written. The need and novelty of the methodology are clearly presented, and the illustration of the use of methodology on realistic instances clearly demonstrates how it works and improves on the existing methodologies for learning in non-stationary data regime.

**Weaknesses:**

Some parts can be improved -
1) Section 2 - Simplification of equation (6) to (9) requires some explanation [page -5].
2) Section 3 - Some intuition behind the objective (11) [page - 6] would be good to understand it better (especially the second term in that objective)
3) Some discussion on how does the methodology perform relative to the degree of non-stationarity in the data?
4) Some minor typos in line 109, 154, 190

**Questions:**

How does the methodology perform relative to the degree of non-stationarity in the data? Specifically, to what extent does this methodology remain effective under varying levels of non-stationarity?

**Limitations:**

Authors have adequately addressed the limitations of their work.

---

> ### Author Rebuttal · Authors · 2024-08-06
>
> We thank the reviewer 6GY1 for their positive feedback. Please find our answer below.
>
> > Section 2 - Simplification of equation (6) to (9) requires some explanation [page -5]
>
> The equation 9 is obtained from 6 as follows. First, we linearise the log-likelihood function $\log p(y_{t+1} | x_{t+1}, \theta)$ around $\mu_t$ which equals to $\log p(y_{t+1} | x_{t+1}, \theta) \sim log p(y_{t+1} | x_{t+1}, \mu_t) + g_{t+1}^T (\theta - \mu_t)$. Then, we notice that $p(y_{t+1} | x_{t+1}, \theta) = \exp^{\log p(y_{t+1} | x_{t+1}, \theta)}$ which means that $p(y_{t+1} | x_{t+1}, \theta) \sim p(y_{t+1} | x_{t+1}, \mu_t) \exp^{-g_{t+1}^T \mu_t} \exp^{g_{t+1}^T \theta}$. Then, since $q_{t}^{t+1}(\theta|S_t,\Gamma_{t-1},\gamma_t)$ is a Gaussian, we can write compute the integral in a closed form. After that, we keep only the terms which depend on $\gamma_{t}$, since we only are interested in the optimization of $\gamma_t$.
>
> We will add a full derivation of this expression in the Appendix.
>
> > Section 3 - Some intuition behind the objective (11) [page - 6] would be good to understand it better (especially the second term in that objective)
>
> The equation in objective 11 is negative Evidence Lower Bound (ELBO) on approximate predictive log likelihood eq 6, if all $\lambda_i = 1$. It could be derived by considering the integral in the right-hand side of eq. 6, dividing and multiplying it by $q(\theta)$ and applying Jensen Inequality. The fact that we introduce $\lambda_i \neq 1$ means that we introduce a temperature parameter (see [1]) on the prior for each dimension $i$. It was shown empirically [1] that using a temperature in the prior leads to better empirical results. We will add more explicit explanations of how to derive the objective 11 and will add full derivation in the Appendix.
>
>
> > How does the methodology perform relative to the degree of non-stationarity in the data? Specifically, to what extent does this methodology remain effective under varying levels of non-stationarity?
>
> Our experiments in Figure 3 and Figure 4 partially answer this question. The difference between Data Efficient and Memorization regimes in Figure 3 is the amount of epochs given to a task – 70 epochs for Data Efficient and 400 for Memorization. There is more non-stationarity in the Data-Efficient setting since it changes more frequently and our methodology remains more effective than baselines.
>
> In general, we expect our method to be very helpful in scenarios when there are relatively long stationary phases which are succeeded by large changes of the optimal parameters. Moreover, given that we can define $\gamma_t$ per parameter or per layer, we can have a combination of these scenarios -- some parameters stay stationary whereas other change significantly. In general, such scenario would occur when we have stationary segments which are followed by non-stationary ones. In case, when the non-stationarity is very strong and the data distribution changes at every step, we think that it is less likely that our method will bring an additional benefit over SGD since in this setting, SGD will be constantly refreshed by the noise from the data. SGD, however, will struggle when there is a large change after a long stationary phase. Compared to reset-type algorithms, our method is much more adaptive, and relearns the parameters only when it "has to" (i.e., when the data changes sufficiently).
>
> We see a partial support for this claim in our RL experiments in Figure 4, where as we become more off-policy, we see much more benefit of our method over standard learning approach. Moreover, we also see a benefit of our method over baseline in a more on-policy regime in Figure 4, left, top on Humanoid environment. Since this environment is hard to solve, the baseline agent will see a lot of similar data initially and therefore the corresponding RL algorithm could overfit. Constantly refreshing the parameters is beneficial here. On the other hand, if we study Figure 4, left, bottom which shows performance in Hopper environment, we can see that Soft Resets is less effective. In this setting, since the problem is easy, the baseline method sees a lot of different data constantly and is less prone to overfit.
>
> > Some minor typos in line 109, 154, 190
>
> Thank you for the catch, we will make the adjustments.
>
> **References**:
>
> [1] How Good is the Bayes Posterior in Deep Neural Networks Really?, Florian Wenzel, Kevin Roth, Bastiaan S. Veeling, Jakub Świątkowski, Linh Tran, Stephan Mandt, Jasper Snoek, Tim Salimans, Rodolphe Jenatton, Sebastian Nowozin, 2020

---

### Official Review · Reviewer_Hmj6 · 2024-07-18

**Soundness:** 3
**Presentation:** 2
**Contribution:** 3
**Rating:** 4
**Confidence:** 4

**Summary:**

This work focuses on the problem of plasticity loss in non-stationary problems. This work proposes a new solution which adaptively drifts the parameters toward the initial distribution. The proposed solution is a form of soft resetting and can be seen as a meta version of L2-init where the degree of drift towards initialization is learned from data.

**Strengths:**

The proposed solution is novel. Current soft resetting methods reset the parameters of the model at a constant rate, irrespective of the degree of non-stationarity. The proposed solution introduces adaptive soft resetting, where the degree of resetting is calculated based on the degree of non-stationarity. This adaptiveness allows the solution to adapt faster and retain more prior knowledge. The experimental results show the effectiveness of the proposed solution.

**Weaknesses:**

Although the proposed solution is novel and effective, the paper has many weaknesses:
1. Hyperparameter sensitivity. Given that the proposed solutions introduce many new hyperparameters and the authors say in line 207 that one of the solutions is sensitive to the choice of hyperparameter, the paper needs to contain hyperparameter sensitivity curves for all new hyperparameters.
2. Computational cost. The proposed solutions seem computationally expensive. It would be good to include the wall of each algorithm beside one of the figures, and Figure 3 might be best.
3. Limited performance improvement. The primary purpose of high replay-ratio is to improve the sample efficiency of algorithms. However, results in Figure 4 show that the proposed solution does not utilize a high replay-ratio. Its sample efficiency at replay ratio 1 is the same as its sample efficiency at replay ratio 128.

**Questions:**

See Weaknesses

**Limitations:**

Computational cost and hyperparameter sensitivity need to be discussed in the conclusion of the main paper.

---

> ### Author Rebuttal · Authors · 2024-08-06
>
> We thank the reviewer Hmj6 for their feedback. Please find our detailed answer below.
>
> > Hyperparameter sensitivity. Given that the proposed solutions introduce many new hyperparameters and the authors say in line 207 that one of the solutions is sensitive to the choice of hyperparameter, the paper needs to contain hyperparameter sensitivity curves for all new hyperparameters.
>
> We have provided the answer about hyperparameters sensitivity above in the common answer for all the reviewers.
>
> > Computational cost. The proposed solutions seem computationally expensive. It would be good to include the wall of each algorithm beside one of the figures, and Figure 3 might be best.
>
> We have provided the answer about computational complexity above in the common answer for all the reviewers.
>
> > Limited performance improvement. The primary purpose of high replay-ratio is to improve the sample efficiency of algorithms. However, results in Figure 4 show that the proposed solution does not utilize a high replay-ratio. Its sample efficiency at replay ratio 1 is the same as its sample efficiency at replay ratio 128.
>
> Our choice of Reinforcement Learning experiment is motivated by the Primacy Bias [1] paper, where a significant plasticity loss occurs and where the benefits of parameter resets were observed. In this setting, it was shown that as replay ratio increases, the performance of the algorithm (Soft Actor Critic) could significantly degrade and hard-resetting parameters once in a while, can significantly improve its performance. With this in mind, our Figure 4 demonstrates that Soft Reset can in fact be an even more effective strategy when off-policy ratio increases – it can learn when to reset parameters and by what amount, compared to Hard Resets which require setting these parameters manually.
>
> We are happy to take further questions on any of the above matters, and we hope that we have been able to address your concerns. If we have done so, we would be grateful if you would consider increasing your review score.
>
> **References**:
>
> [1] The Primacy Bias in Deep Reinforcement Learning, Evgenii Nikishin, Max Schwarzer, Pierluca D'Oro, Pierre-Luc Bacon, Aaron Courville, 2022

---

### Author Rebuttal · Authors · 2024-08-06

Dear reviewers, thank you for your feedback. In this section we provide answers to recurrent points from some of you.

# Computational Complexity

The following tables will be added to the Appendix together with a reference to this appendix at the end of Section 3.

**Notations**:

* P - number of parameters of Neural Network (NN)
* L - number of layers.
* $O(S)$ - cost of backwards pass of SGD
* $K$ - number of Monte Carlo (MC) samples in eq.7
* $J$ - number of iterations in eq. 7
* $I$ - number of parameter updates for proximal methods in eq.14 and eq.18
* $M$ - number of MC samples for Bayesian method in eq. 12


| Method                                      | Comp. cost.  | Memory           |
|----------------------------------------------|--------------|------------------|
| SGD                                          | O(S)         | O(P)             |
| Soft resets $\gamma$ p. layer                  | O(JKS + S)    | O(L+(K+1)P)     |
| Soft resets $\gamma$ p. param.                 | O(JKS + S)    | O(P+(K+1)P)     |
| Soft resets $\gamma$ p. layer + proximal (I iters) | O(JKS + I S)  | O(L+(K+1)P)     |
| Soft resets $\gamma$ p. param. + proximal (I iters) | O(JKS + I S)  | O(P+(K+1)P)     |
| Bayesian Soft Reset Proximal (I Iter) $\gamma$ p.layer | O(JKS + 2M I S)| P(L+(K+2)P)     |
| Bayesian Soft Reset Proximal (I Iter) $\gamma$ p.param | O(JKS + 2M I S)| P(P+(K+2)P)     |
|

The table above denotes the theoretical cost of each of the methods. In practice, we use $J=1$ for Soft Resets and $J=10$ for proximal methods. The table below quantifies the exact cost for methods from Figure 3.

| Method                                      | Comp. cost.  | Memory           |
|----------------------------------------------|--------------|------------------|
| SGD                                          | O(S)         | O(P)             |
| Soft resets                                  | O(2S)         | O(L+2P)          |
| Soft resets more compute                     | O(11S)        | O(L+2P)          |
| Soft resets proximal                         | O(10S + 10S) | O(L+2P)          |
| Bayesian Soft Reset Proximal                 | O(10S + 20S) | P(L+3P)          |
| Bayesian Soft Reset Proximal $\gamma$ p.param | O(10S + 20S) | P(P+3P)          |
|

From Figure 3, we see that it is beneficial to spend more compute on optimizing $\gamma$ and NN parameters. However, even the cheapest Soft Resets leads to a good performance.

In Reinforcement Learning (RL) experiment, we do **one** update on $\gamma$ after each new chunk of fresh data. We do $G$ updates on NN parameters with a cost $O(S)$ each. The table below denotes the complexities.

| Method        | Complexity  | Memory       |
|---------------|-------------|--------------|
| SAC           | O(G S)      | O(P)         |
| Soft Reset    | O(S + G S)  | O(L + 2 * P) |
|

Soft Reset is marginally more expensive than SAC in RL but leads to a better performance (see Figure 4) in a highly off-policy regime.

# Hyperparameters sensitivity

We study the sensitivity of **Soft Resets** where $\gamma$ is defined per layer.

**Fixed parameters:**

* Number of MC samples $K=1$ and $M=1$
* Learning rate for parameters was tuned prior to that and equals to $\alpha=0.1$.

**Sensitivity parameters** (see Algorithm 1 for precise definitions):

* $\eta_{\gamma}$ - learning rate for the drift model
* $f$ - initial prior standard deviation rescaling, i.e., $\sigma^l_0 = f \frac{1}{\sqrt{H}}$ where $H$ is the width of the layer $l$
* $s$ - posterior scaling, i.e., $\sigma^l_t = s * \sigma^l_0$.

We provide the plots for the sensitivity analysis of Soft Reset on MNIST (data efficient) in the attached pdf. On top of that, we conduct the sensitivity analysis of L2 Init [1] and Shrink&Perturb [2] methods. The X-axis of each plot denotes one of the studied hyperparameters, whereas Y-Axis is the average performance across all the tasks (see Experiments section for tasks definition). The standard deviation is reported over 3 random seeds. A color indicates a second hyperparameter which is studied, if available. In the title of each plot, we write hyperparameters which are fixed.

**Takeaways**

The most important parameter is the learning rate $\eta_{\gamma}$ of the drift model. For each method, there exists a good value of this parameter and performance is sensitive to it. This makes sense since this parameter directly impacts how we learn the drift model.

The performance of Soft Resets is robust with respect to the posterior standard deviation scaling $s$ parameter as long as it is $s\geq0.5$. For $s<0.5$, the performance degrades. This parameter is defined from $\sigma_{posterior} = s \sigma_{prior}$ and effects relative increase in learning rate (see eq. 13 and eq. 17). This increase is given by $1/(\gamma^2 + (1-\gamma^2)/s^2))$, which could be ill-behaved for small $s$.

We also study the sensitivity of the baseline methods. We find that L2 Init is very sensitive to the parameter $\lambda$ which is a penalty on $||\theta-\theta_0||^2$ term. In fact, Figure 2, left shows that there is only one good value of this parameter which works. Shrink\&Perturb is very sensitive to the shrink parameter ($\lambda$). Similar to L2 Init, there is only one value which works, $0.9999$ while values $0.999$ and $0.99999$ lead to bad performance. This method however, is not very sensitive to the perturb parameter $\sigma$ provided that $\sigma \leq 0.001$.

Compared to the baselines, our method is more robust to the hyperparameters choice.

We also conduct sensitivity analysis for other variants of the method, but due to space constraints, will only include the results in the camera ready version. The take-aways are similar.

**References:**

[1] Maintaining Plasticity in Continual Learning via Regenerative Regularization, Saurabh Kumar, Henrik Marklund, Benjamin Van Roy, 2023

[2] On Warm-Starting Neural Network Training, Jordan T. Ash, Ryan P. Adams, 2020

---

### Decision · Program_Chairs · 2024-09-25

**Decision:**

Accept (poster)

**Comment:**

This paper proposes to adapt to non-stationarity in neural network training using an Ornstein-Uhlenbeck process with an adaptive drift parameter. The proposed method estimates the drift parameter online and allows for automatic soft resets of parameters when needed. The approach is evaluated on plasticity benchmarks and off-policy reinforcement learning tasks. The reviewers all agree that the main idea is interesting and novel.

However, the significance of the contributions are not clearly demonstrated due to the lack of clarity in the submission. In its current form, even the most engaged readers have some difficulty grasping the core technical ideas, e.g., see the discussion thread with Reviewer 4koq.  In the rebuttal process, the authors commit to resolving several of these issues—I will not repeat them here—which will be critical to the paper having broad impact. In particular, I strongly recommend adding a candid discussion on the computational costs of the proposed approach.